

# Applicability and effectiveness of structural measures for subsidence (risk) reduction in urban areas

Nicoletta Nappo[1,2], Mandy Korff[1,2]

[1]Department of Civil Engineering and Geosciences, Technical University of Delft (TU Delft), Mekelweg 5, 2628 CD Delft, Netherlands
[2]Deltares, Boussinesqweg 1, 2629 HV Delft, Netherlands

*Correspondence to*: Nicoletta Nappo (n.nappo@tudelft.nl ; nicoletta.nappo@deltares.nl)

**Abstract.** Managing subsidence and its impacts on cities in coastal and delta areas is a global challenge that requires comprehensive risk reduction policies, including both mitigation and prevention strategies. Urban areas often lack systematic methodologies for determining appropriate countermeasures. This paper proposes a twofold strategy for selecting subsidence reduction measures in urban areas – which refer to structural (i.e., technical) measures to prevent and mitigate subsidence and its physical consequences - based on their applicability and performance. The Question-and-Response (Q&R) system serves as a decision tree to identify suitable subsidence countermeasures based on their applicability to specific cases. Four indicators of effectiveness – i.e., reduction potential, operational reliability, negative impact and service life – are then used to assess the performance of subsidence reduction measures. The proposed procedure was applied to 49 cases derived from a review of 52 scientific publications and additional expert sessions and surveys involving five academic scholars and 13 experts. Also, the method was applied to examples from Shanghai (China), Jakarta (Indonesia) and San Joaquin Valley (USA, California). The strategies proposed in this paper proved suitable for an initial screening of subsidence reduction measures applicable in different urban areas, after which a site-specific assessment can follow. Furthermore, this study shows the need to collect and share experiences in evaluating the performance of subsidence reduction measures more systematically, and gives a first framework to do so.

## 1 Introduction

Mexico City (Mexico), Jakarta (Indonesia), Bangkok (Thailand), Venice (Italy), New Orleans (Louisiana, USA), Lagos (Nigeria), Hokkaido (Japan), Shanghai (China) and Gouda (Netherlands) are examples of cities affected by subsidence (Bagheri-Gavkosh et al., 2021; Bucx et al., 2015; Davydzenka et al., 2023; Dinar et al., 2021; Erkens et al., 2015; Herrera-García et al., 2021; Hutabarat & Ilyas, 2017; Pedretti et al., 2024; Poland, 1984). The sinking rates in these cities span from few millimetres (for example in Gouda) to tens of centimetres (for example in Jakarta) causing socio-economic distresses and environmental and structural damages (Erkens et al., 2015). The drivers of subsidence are generally distinguished in natural and anthropogenic, although their combination is often the cause of negative impacts in cities (Galloway & Burbey, 2011). Natural causes typically include consolidation of compressible soils, shrinking and swelling of cohesive soils , decomposition



of organic soils, groundwater discharge, karst and tectonic processes (Gambolati and Teatini, 2021; Poland, 1984). Groundwater, gas or oil extraction, mining, underground excavations, urban sprawl and construction loading are anthropogenic factors causing or exacerbating subsidence processes (Gambolati and Teatini, 2021; Poland, 1984). Moreover, the combination of subsidence with sea-level rise and climate changes increases the exposure of cities to additional risks, such as flooding

(Herrera-García et al., 2021).

Unlikely other geological or geophysical hazards with immediate disastrous impacts (e.g., earthquakes, landslides), subsidence is a relatively slow process with moderate intensity that can take decades to turn into a disaster (UNDRR, 2024). For this reason, subsidence is often unnoticed and not acknowledged as a disaster, and its physical, socio-economic and environmental impacts in urban areas are not perceived as a potential catastrophe (Bucx et al., 2015; Erkens et al., 2015; Kok and Costa,

2021). Nevertheless, small-to-large scale subsidence can cause costly short-to-long term negative effects to cities that deserve proper (risk) management and reduction policies (Herrera-García et al., 2021). Several authors (Bucx et al., 2015; Department of Regional NSW, 2023; Erkens and Stouthamer, 2020; Jin et al., 2024; Kok and Costa, 2021; Peduto et al., 2015; Piper, 2021; Sendai Framework for DRR, 2015) proposed frameworks for subsidence (risk) management, outlining four primary steps:

1) Problem analysis. This involves data collection and analysis, determination of subsidence causes, damage assessment, and

(inverse) predictive modelling.

2) Planning. This step encompasses scenario construction, vulnerability and risk assessment, cost-benefit analysis, forecasting, decision support systems, proposing innovative (alternative) solutions, exchanging of knowledge and best practices, and selection of mitigation and prevention measures.

3) Implementation. This involves installing monitoring systems, starting pilot projects, and implementing mitigation and

prevention measures.

4) Evaluation. The final step is dedicated to the assessment of the management cycle and outlook.

Most of the research activities reported in literature (63%) focus on measuring and monitoring subsidence in urban areas using ground-based (e.g., levelling, GPS, extensometers) and remote sensing techniques (as InSAR and LiDAR - e.g., Ezquerro et al., 2020; Herrera et al., 2010; Ikuemonisan et al., 2021; López-Quiroz et al., 2009; Nappo et al., 2021; Peduto et al., 2019);

30% reports on modelling and forecasting; while only 7% provides examples of cities where mitigation and prevention measures are applied (Scopus, 2024).

Technical interventions are commonly employed to protect major cities from subsidence; however, a systematic and objective method for selecting suitable solutions has not yet been established. Additionally, because of the diversity of mitigation and prevention methods, subsidence characteristics, impacted (infra)structures and societies, evaluating the short- and long-term

performance of subsidence countermeasures remains challenging. In this perspective, this paper aims at bridging this gap by proposing a twofold strategy to select mitigation and prevention measures based on their applicability and performance. First, a system of Question-and-Response (Q&R) is proposed to identify suitable subsidence mitigation and prevention measures tailored to the specific requirements of each case. Then, by leveraging methods used to assess the effectiveness of mitigation measures against earthquakes, snow avalanches, landslides and floods (Bründl et al., 2016; Hudson et al., 2014; Januriyadi et



al., 2020; Margreth and Romang, 2010), this paper introduces four indicators to evaluate the effectiveness of selected subsidence countermeasures. This paper focusses on structural (i.e., technical) measures to counteract subsidence risk in urban areas, addressing both ground settlements and the resulting physical consequences (i.e., damage) to structures. With few adjustments, the proposed methodology could be adapted for non-structural (i.e., non-technical) measures, socio-economic and environmental effects or subsidence countermeasures in rural areas; this however is not the aim of this paper.

After this introduction, the paper is structured as follows: Section 2 recalls the definitions of reduction, mitigation, prevention and adaptation used in this study; Section 3 presents the collected data; Section 4 introduces the Q&R system and the indicators of effectiveness; Section 5 applies the methodology to selected cases and analyses the obtained results; Section 6 and Section 7 respectively discuss and conclude this paper. A brief description of measures to counteract subsidence and its physical consequences in urban areas is provided in Appendix A.

## 2 Definitions


The definitions of terms given hereafter are based on the United Nations Multilingual Terminology Database (UNTERM, 2024) and the Sendai Framework Terminology on Disaster Risk Reduction (UNDRR, 2024). These definitions strictly refer to subsidence risk management; therefore, some of them may differ in other contexts, such as in climate change policies and civil structural engineering.

• *Reduction*. Strategies to decrease or remove the risk of subsidence by acting on the predisposing factors, magnitude, intensity or frequency of subsidence, or on the vulnerability and exposure of urban areas affected by it. Subsidence reduction measures encompass both mitigation and prevention measures.

• *Mitigation*. Structural and non-structural measures taken to minimise subsidence and its adverse impacts (e.g., damages) that cannot be entirely prevented. In urban areas, mitigation examples include repairing cracks in buildings following ground settlements or re-injecting fluids into aquifers after extraction.

• *Prevention*. Structural and non-structural measures taken to entirely avoid subsidence and its adverse impacts (e.g., damages) and to avert cascading effects such as sinkholes or increased flood risk. In urban areas, prevention examples
include employing deep foundations for buildings in soft soils or enhancing soil strength before construction.

• *Adaptation*. Adjusting to the adverse impacts of subsidence or its evolving conditions that cannot be avoided or modified. This term is mainly used in the field of climate change. For subsidence in urban areas, it refers to non-structural measures.


• *Structural and non-structural measures*. Set of technical interventions and non-technical strategies employed to cope with new or existing subsidence and its (potential) disastrous consequences. Structural interventions involve hazard-resistant physical structures and engineering techniques to withstand the physical impacts of subsidence. Non-structural measures include laws, regulations, alternative urban planning, public awareness initiatives, and
environmental and social policies. The terms "structural and non-structural measures" in subsidence risk management differ from their usage in civil and structural engineering.



Other terms such as "remedial", "reparative", "precautionary", "protective" or "compensatory" measures to "control or arrest" subsidence and its physical consequences can be found in literature (Nutalaya et al., 1996; Poland, 1984; Singh and Dhar, 1997; Stouthamer et al., 2020; Zektser et al., 2005), referring to what here is defined as "mitigation" and "prevention" measures.

It should be noted that, in this paper, the terms "subsidence countermeasures" and "subsidence reduction measures" are used interchangeably. Both terms refer to mitigation and prevention measures employed in urban areas to contrast subsidence and its physical consequences on (infra)structures.

**3 Data collection**

Scientific papers and technical articles were retrieved from publication databases and search engines (e.g., Google Scholar, Scopus). A set of 52 publications was selected for the purpose of this study because they describe cases where structural measures are used for contrasting subsidence and damage to structures in urban areas. Additionally, two expert sessions and surveys were organized by the authors to gather experiences from five academic scholars and 13 experts on subsidence mitigation and prevention.

Table 1 lists the selected publications and the cases discussed during the expert sessions and surveys, detailing the location, cause of subsidence, average settlement rate, geology and subsidence countermeasures for each case study.

**Table 1. List of publications and cases discussed during expert sessions and surveys that, to the authors' knowledge, document instances where structural (i.e., technical) measures have been employed to contrast subsidence and damage to structures in urban areas.**

| Reference | Location (Country, city) | Cause of subsidence | Average rate of subsidence | Geology | Subsidence reduction measures |
|---|---|---|---|---|---|
| Abidin et al., 2015 | Indonesia, Jakarta | Groundwater extraction, construction loading | 3-10 cm/year | Alluvial deposits | Aquifer recharge |
| Akbar et al., 2019 | Indonesia, Semarang | Groundwater extraction, construction loading | 6-7 cm/year; 14-19 cm/year in some areas | Alluvial deposits | Retention pond, elevation of linear infrastructures |
| Alferink and Cordóva, 2017 | Netherlands, Groningen Province | Gas extraction, seismic activity | 0.3-0.5 cm/year | Sand, clay | Flexible connections to underground infrastructures |





| Al-Zabedy and Al-Kifae, 2020 | Iraq | Karst erosion | - | Gypsum | Improved foundations, soil injections, dynamic compaction of soil |
|---|---|---|---|---|---|
| Andreas et al., 2018 | Indonesia, Jakarta | Groundwater extraction, construction loading | 1-10 cm/year; 20-26 cm/year in some areas | Sand, silts and clay | Building jacking, elevation of linear infrastructures, structure relocation |
| | Indonesia, Semarang | Groundwater extraction, construction loading | 6-7 cm/year; 14-19 cm/year in some areas | Alluvial deposits | Building jacking, elevation of linear infrastructures |
| Andriani et al., 2021 | Indonesia, Tanjung Api-Api | Soil compaction and oxidation, groundwater extraction | 5 cm/year | Peat, clay | Infiltration well, retention pond, accelerate soil consolidation, elevation of linear infrastructures, lightweight construction materials |
| Basak and Chowdhury, 2021 | Netherlands, Maasbommel | Shrink and swell, groundwater extraction, construction loading | < 0.1 cm/year | Clay | Floating and amphibious housing |
| | Bangladesh, Dhaka | Groundwater extraction | 0.3-2 cm/year | Gravel, sand, silt, clay | Floating and amphibious housing |
| Bell et al., 2002 | USA, Las Vegas, Nevada | Groundwater extraction | 5-6 cm/year | Silt, clay | Aquifer recharge, retention pond |
| Bergado et al., 1993 | Thailand, Bangkok | Groundwater extraction, soil compaction | 10 cm/year | Clay | Accelerate soil consolidation, mechanical soil mixing |
| Brighenti, 1991 | Italy, Abano Terme | Groundwater extraction | 6 cm/year | Marly limestone | Injection well |



| Carreón-Freyre et al., 2010 | Mexico, Itzapalapa, Mexico City | Groundwater extraction, construction loading | 12 cm/year | Clay | Repairing cracks, elevation of linear infrastructures |
|---|---|---|---|---|---|
| Deakin, 2005 | UK, Wiltshire | Shrink and swell | - | Clay | Improved foundations, repairing cracks |
| English et al., 2016; 2021 | USA, New Orleans, Louisiana | Soil compaction | 1 cm/year | Peat | Floating and amphibious housing |
| English et al., 2021 | Netherlands, Maasbommel | Shrink and swell, groundwater extraction, construction loading | < 0.1 cm/year | Clay | Floating and amphibious housing |
| Galloway and Riley, 1999 | USA, San Joaquin Valley, California | Groundwater extraction, soil compaction | 2.7-22 cm/year | Clay | Retention pond, injection well |
| Gambolati et al., 2005 | USA, Wilmington, California | Oil extraction | 2.25 cm/year | Sand, silt | Injection well |
| | Italy, Venice | Groundwater extraction, soil oxidation, construction loading | 0.2 cm/year | Alluvial deposits | Injection well |
| Gutiérrez and Cooper, 2002 | Spain, Calatayud | Karst erosion | 2 cm/year | Gypsum | Flexible connections to underground infrastructures, improved foundations |
| Hamidi et al., 2011 | UAE, Abu Dhabi | Groundwater extraction | - | Silty sand | Dynamic compaction of soil |

| | | | | | |
|---|---|---|---|---|---|
| Han, 2003 | China, Beijing | Groundwater extraction | 5 cm/year | Silty clay | Aquifer recharge, retention pond |
| | China, Luo River | | - | Alluvial deposits | Aquifer recharge |
| | China, Qingdao | | 3 cm/year | Alluvial deposits | Aquifer recharge |
| | China, Shanghai | | 6 cm/year | Sand, clay | Injection well |
| | China, Tianjin | | 3 cm/year | Alluvial deposits | Injection well |
| Huang et al., 2015 | China, Shanghai | Groundwater extraction, construction loading | 6 cm/year | Sand, clay | Injection well |
| Jha et al., 2009 | Japan, Kochi Prefecture | Groundwater extraction | - | Silty sand and gravel | Aquifer recharge, retention pond, exfiltration sewer |
| Kohlnhofer, 1992 | Norway | Soil compaction | - | Peat | Lightweight construction material |
| | USA, Pickford, Michigan | Soil compaction | - | Silty clay | Lightweight construction material |
| Kok and Hommes-Slag, 2020 | Netherlands, Gouda | Organic soil oxidation, groundwater extraction, construction loading | 0.3 cm/year | Peat | Compartmentalization, elevation of linear infrastructures, improved foundations, lightweight construction materials |
| Li et al., 2021 | China, Shanghai, Nanpu bridge | Groundwater extraction | 5 cm/year | Silt, sand | Injection well |
| Liang et al., 2015 | China, Ningbo Port | Soft soil compaction | 5 cm/year | Clay, fly ash and silty sand | Dynamic compaction of soil |
| Lixin et al., 2022 | China, Tianjin | Groundwater extraction | 7 cm/year | Alluvial deposits | Retention pond |
| Luo et al., 2019 | USA | Coal mining | - | - | Repairing cracks |



| McBean et al., 2019 | China, Beijing | Groundwater extraction | 5 cm/year | Silty clay | Exfiltration sewer |
|---|---|---|---|---|---|
| Nutalaya et al., 1996 | Thailand, Bangkok | Soil consolidation, construction loading, groundwater extraction | 10 cm/year | Clay, sand | Aquifer recharge |
| Ovando-Shelley et al., 2013 | Mexico, Mexico City | Groundwater extraction | 7-10 cm/year | Clay | Improved foundations |
| Pacheco-Martínez et al., 2013 | Mexico, Aguascalientes | Groundwater extraction, construction loading | 7.2 cm/year | Sand and gravel with silt and clay | Aquifer recharge, demolition of unsafe buildings |
| Paukstys et al., 1999 | Lithuania, Birai | Karst erosion | - | Gypsum | Flexible connections to underground infrastructures |
| | UK, Ripon | Karst erosion | - | Gypsum | Flexible connections to underground infrastructures |
| Phien-Wej et al., 1998 | Thailand, Bangkok | Groundwater extraction | 10 cm/year | Sand, gravel and clay | Injection well |
| Poland, 1984 | China, Shanghai | Groundwater extraction | 6 cm/year | Sand and clay | Injection well |
| | UK, Cheshire | Salt mining | 3.38 cm/year | Marl, sandstone | Elevation of linear infrastructures, improved foundations |
| | Japan, Tokyo | Groundwater extraction | 7.6 cm/year; 24 cm/year in some areas | Alluvial deposits | Retention pond, aquifer recharge |



| | South Africa, Far West Rand, Johannesburg | Gold mining | 56 cm/year | Dolomite and unconsolidated deposits | Injection well |
|---|---|---|---|---|---|
| | USA, Alabama | Mining, karst erosion | 49 cm/year | Carbonate rocks | Elevation of linear infrastructures, accelerate soil consolidation |
| | USA, Santa Clara Valley | Groundwater extraction | 7.8 cm/year | Alluvial deposits | Retention pond, aquifer recharge, permeable pavement |
| Pötz and Bleuzé, 2009 | Netherlands, Maasbommel | Shrink and swell, groundwater extraction, construction loading | < 0.1 cm/year | Clay | Floating and amphibious hosing |
| Pramono, 2021 | Indonesia, Semarang | Groundwater extraction, construction loading | 6-13 cm/year | Alluvial deposits | Retention pond |
| | Indonesia, Jakarta | Groundwater extraction, construction loading | 11-13 cm/year | Sand, silts and clay | Retention pond, exfiltration sewer |
| Ritzema, 2015 | Netherlands, Maasbommel | Shrink and swell, groundwater extraction, construction loading | < 0.1 cm/year | Clay | Accelerate soil consolidation, flexible connections to underground infrastructures, floating and amphibious housing, improved foundations, lightweight construction materials |





| | | | | | |
|---|---|---|---|---|---|
| Saputra et al., 2017, 2019 | Indonesia, Jakarta | Groundwater extraction, construction loading | 1-15 cm/ year; 25-28 cm/ year in some areas | Sand, silts and clay | Building jacking, infiltration well |
| | Indonesia, Semarang | Groundwater extraction, construction loading | 8-13.5 cm/year | Alluvial deposits | Building jacking, lightweight construction materials |
| Shen et al., 2019 | Taiwan, Lukang district | Liquefaction | - | Sand | Dynamic compaction of soil |
| Shi et al., 2016 | China, Shanghai | Groundwater extraction | 6 cm/year | Sand and clay | Injection well |
| Sneed and Brandt, 2020 | USA, Coachella Valley, California | Groundwater extraction | 10 cm/year | Gravel sand, silt and clay | Aquifer recharge, retention pond |
| Szucs et al., 2009 | Hungary, Debrecen | Groundwater extraction | 0.8 cm/year | Sand | Aquifer recharge, retention pond, infiltration well |
| Tang et al., 2022 | China, Taiyuan basin | Groundwater extraction | 8 cm/year | Soft soil and sand | Injection well |
| Testa, 1991 | USA, Wilmington Area, Los Angeles, California | Oil and groundwater extraction | 36-45 cm/year | Sand and gravel alternated with silt and clay | Injection well |
| Ting et al., 2020 | Taiwan, Pingtung Plain | Groundwater extraction | 1.6 cm/year | Alluvial deposits | Aquifer recharge, retention pond |
| Wu et al., 2020 | China, Shanghai | Groundwater extraction | 6 cm/year | Sand and clay | Injections well |
| Xuan et al., 2015 | China, Anhui Province | Coal mining | 10 cm/year | Silt | Soil injections |
| Yang et al., 2020 | China, Shanghai | Groundwater extraction, | 6 cm/year | Sand and clay | Injection well |



| | | construction loading | | | |
|---|---|---|---|---|---|
| Ye et al., 2016 | China, Shanghai | Groundwater extraction | 6 cm/year | Sand and clay | Injection well |
| Zektser et al., 2005 | USA, San Francisco, California | Groundwater extraction | 0.2 cm/year | Alluvial deposits | Retention pond |
| | USA, Redwook Creek, California | Groundwater extraction | - | Alluvial deposits | Retention pond |
| Expert sessions and survey | Netherlands, Amsterdam | Soil compaction, shrink and swell, building loading | 0.1-0.3 cm/year | Clay, sand | Accelerate soil consolidation, injection well |
| | Netherlands, Rotterdam | Soil compaction, groundwater extraction, construction loading | 0.2-0.3 cm/year | Clay, sand | Infiltration well, exfiltration sewer |
| | Netherlands, Woerden | Soil compaction and oxidation, shrink and swell, construction loading, groundwater extraction | 0.1-0.4 cm/year | Clay, peat, sand | Floating and amphibious housing, improved foundations, lightweight construction materials |
| | USA, Houston, Texas | Groundwater extraction | 0.5-2 cm/year | Clay and sand | Aquifer recharge, retention pond |



| USA, New Orleans, Louisiana | Groundwater extraction | 0.6-0.8 cm/year | Peat and clay | Retention pond, exfiltration sewer, building jacking, improved foundations |
|---|---|---|---|---|

A more detailed description of the subsidence countermeasures mentioned in Table 1 is provided in Appendix A.

## 4 Method to select subsidence reduction measures

This section describes the two-step approach proposed in this paper to select subsidence reduction measures in urban areas
based on their applicability and estimated effectiveness. The applicability of subsidence countermeasures is determined via
the Question-and-Response (Q&R) system. Then, four indicators are used to evaluate the performance of subsidence reduction
measures in terms of effectiveness.

### 4.1 Applicability: the Question-and-Response (Q&R) system

Besides a first distinction between structural and non-structural, subsidence reduction measures can be categorized as outlined
in Table 2. These categories derive from a set of questions and responses selected by the authors together with the academic
scholars and experts consulted for this study, and they reflect the key requirements influencing the selection of subsidence
countermeasures in urban areas. By answering these questions, the applicability of each subsidence countermeasure to specific
cases can be assessed. The Q&R system provides stakeholders and decision makers with a tool to rapidly identify (a set of)
suitable subsidence reduction measures that meet the specific requirements of each case.

Depending on the application, location and available information, additional sub-categories (e.g., type of soil/rock, direct and
indirect impacts, involved costs, etc.) can be added to the system, thus reaching a further level of detail. However, to facilitate
a broader comparison among different applications, this paper does not include any sub-category. This decision is based on
the review of worldwide case studies, where the inclusion of sub-categories would hinder the comparability of diverse
applications.

**Table 2. Question-and-Response (Q&R) system serving as a decision tree to identify suitable subsidence reduction measures based on their applicability.**

| Question | Response | Category |
|---|---|---|
| What is the (potential) area of influence of the subsidence reduction measure? | < 100 m$^2$ | *Micro scale* |
| | 100 m² to 1,000 m² | *Small scale* |
| | 1,000 m² to 100,000 m² (0.1 km²) | *Medium scale* |



| | 100,000 m² (0.1 km²) to 1,000,000 m² (1 km²) | *Large scale* |
| --- | --- | --- |
| | > 1,000,000 m² (1 km²) | *Regional scale* |
| What is the primary cause of subsidence in the area? | Consolidation of compressible soil, shrinking and swelling of cohesive soils, decomposition of organic soils, groundwater discharge, karst and tectonic processes | *Natural subsidence* |
| | Fluid extraction, mining, underground excavations, urban sprawl and construction loading | *Anthropogenic subsidence* |
| What is the predominant geology of the area? | Peat, silt, clay, sand, gravel | *Soils* |
| | Limestone, gypsum, etc. | *Rocks* |
| What is the primary objective of the intervention? | Avoid (new or additional) subsidence and its adverse impacts | *Prevention* |
| | Reduce subsidence and its adverse impacts | *Mitigation* |
| What needs to be prevented or mitigated? | Subsidence | *Hazard* |
| | Damage to structures | *Vulnerability & Exposure* |
| What type of urban area is involved? | Existing area | *Rehabilitation* |
| | Expansion area | *New development* |
| Where is the subsidence countermeasure to be applied? | Roads, streets, squares, parks, monuments, schools, parking, etc. | *Public space* |
| | Houses, gardens, shops, etc. | *Private space* |
| What type of intervention is being considered? | Physical structures | *Structural measure* |
| | Laws, regulations, spatial planning | *Non-structural measure* |

### 4.2 Indicators of effectiveness

Once (a set of) suitable subsidence countermeasures is identified for a specific case, their effectiveness can be evaluated using four indicators: reduction potential, operational reliability, negative impact and service life. A subsidence reduction measure is effective when it performs well across all the indicators and it contributes to reducing the (risk of) subsidence and its physical consequences in urban areas.

- **Reduction potential (RP)**. *How much subsidence and its physical consequences can be reduced?* This indicator estimates the percentage of subsidence and damage reduction, and it is ranked as:
    - *High*: RP >= 50%
    - *Medium*: 10% <= RP < 50%
    - *Low*: RP < 10%





- **Operational reliability (OR)**. *Does the subsidence countermeasure perform as intended over time without failure?*
  This indicator reflects the functionality of subsidence reduction measure. If the system reaches or exceeds its limit
  state (i.e., the system fails), the subsidence countermeasure loses its effectiveness and may require (major) restoration
  or replacement to re-establish its functionality. This indicator can be classified as:
    - *Good:* No interventions are needed.
    - *Fair:* Minor interventions are needed.
    - *Bad:* Major interventions are needed.

- **Negative impact (NI)**. *Does the subsidence countermeasure have negative side effects?* This indicator evaluates
  whether a subsidence reduction measure cause any detrimental effect to the surrounding natural and built
  environment. It can be classified as:
    - *Minimal*: No or minimal negative impacts are observed.
    - *Significant:* Notable negative impacts are observed.

- **Service life (SF)**. *What is the (expected) service life of the subsidence countermeasure?* This indicator reflects the
  expected duration for which a subsidence reduction measure is able to contrast subsidence and its physical
  consequences. It can be classified as:
    - *Long*: SF >= 10 years,
    - *Short*: SF < 10 years.

## 5 Application of the proposed approach

This paper analysed 49 cases distributed in 18 Countries, as shown in Fig. 1. The United States of America (USA), China and
The Netherlands are the countries with the highest number of locations where applications of subsidence countermeasures
have been reported. It is worth underlining that the number of cities known to be affected by subsidence differs from the cases
investigated here (see for example Davydzenka et al., 2023; Pedretti et al., 2023).



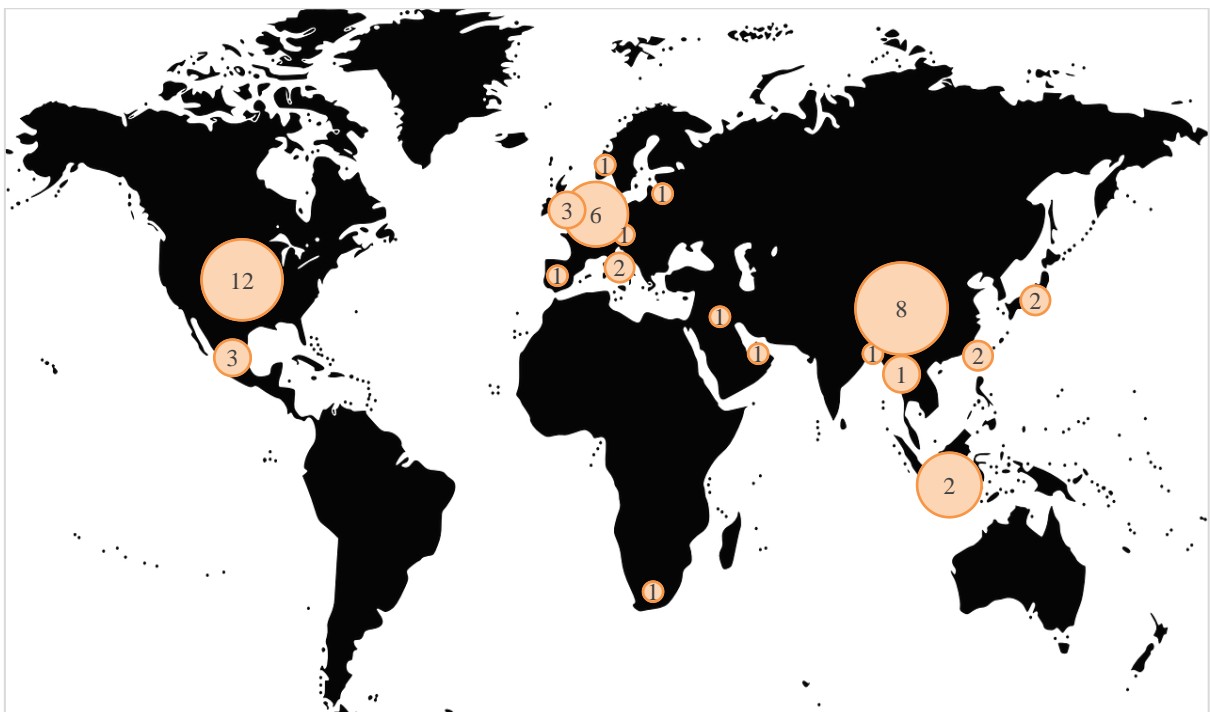

**Figure 1. World map showing the number of cases investigated per country. The size of the bubbles is proportional to the number of scientific papers considered in this study.**

Figure 2 shows that 71% of the 49 investigated cases identify anthropogenic activities as the primary cause of subsidence, while the remaining 29% are attributed to natural causes. Additionally, 32% of the 49 cases has also a secondary cause of subsidence, with 18% of them being anthropogenic and 14% being natural. In 18% of the 49 cases, subsidence is attributed to more than two causes. Groundwater extraction is the most common primary and secondary cause of subsidence, whereas construction loading and soil compaction are mostly identified as secondary or tertiary cause.



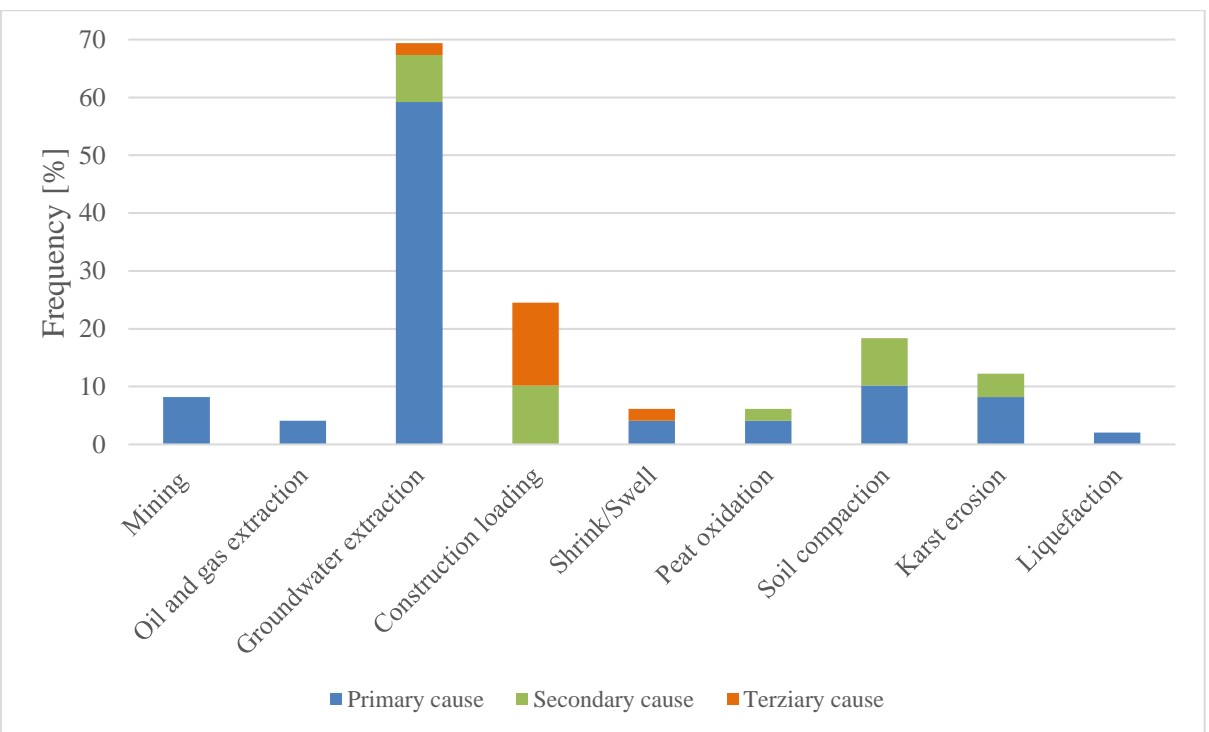

**Figure 2. Frequency of the (anthropogenic and natural) causes of subsidence in the investigated case studies.**

The analysis of the scientific literature, expert sessions and surveys reveals that 84% of the investigated cases are characterized by a geology predominantly composed of soils, while the remaining 16% are primarily composed of rocks (Fig. 3). Among the soil types, clay and sand are the most frequent, representing 26% and 23% of the cases, respectively. A single dominant lithology is observed in 61% of the 49 cases, whereas the remaining 39% exhibit a more complex geological structure with multiple lithologies.





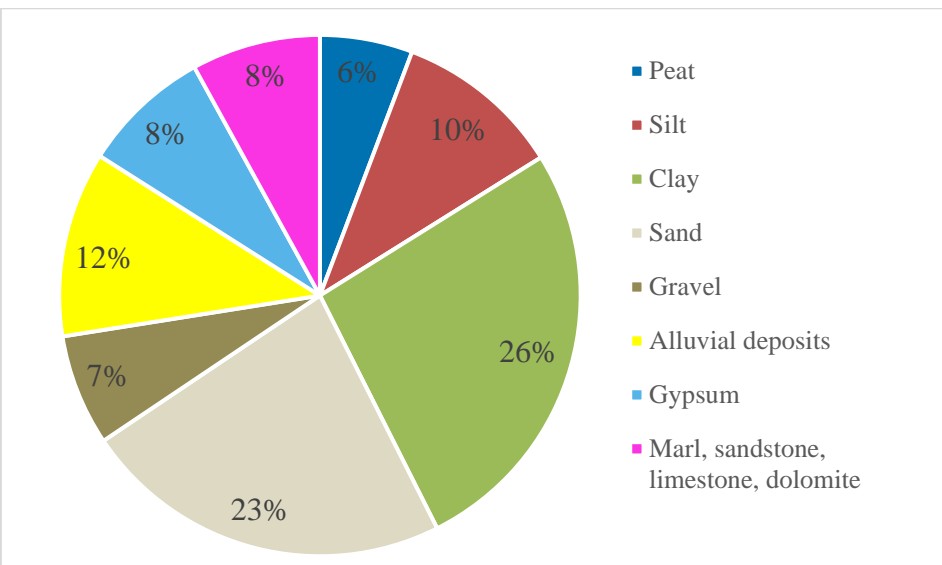

**Figure 3. Distribution of geological types the investigated case studies.**


As for the subsidence reduction measures adopted in the investigated cases, Fig. 4 shows that the majority of the interventions (51%) are related to (ground)water management, followed by construction improvements (39%) and soil improvements (10%). The most frequently employed measures are 'Retention pond' (17%) and 'Aquifer recharge' (14%).





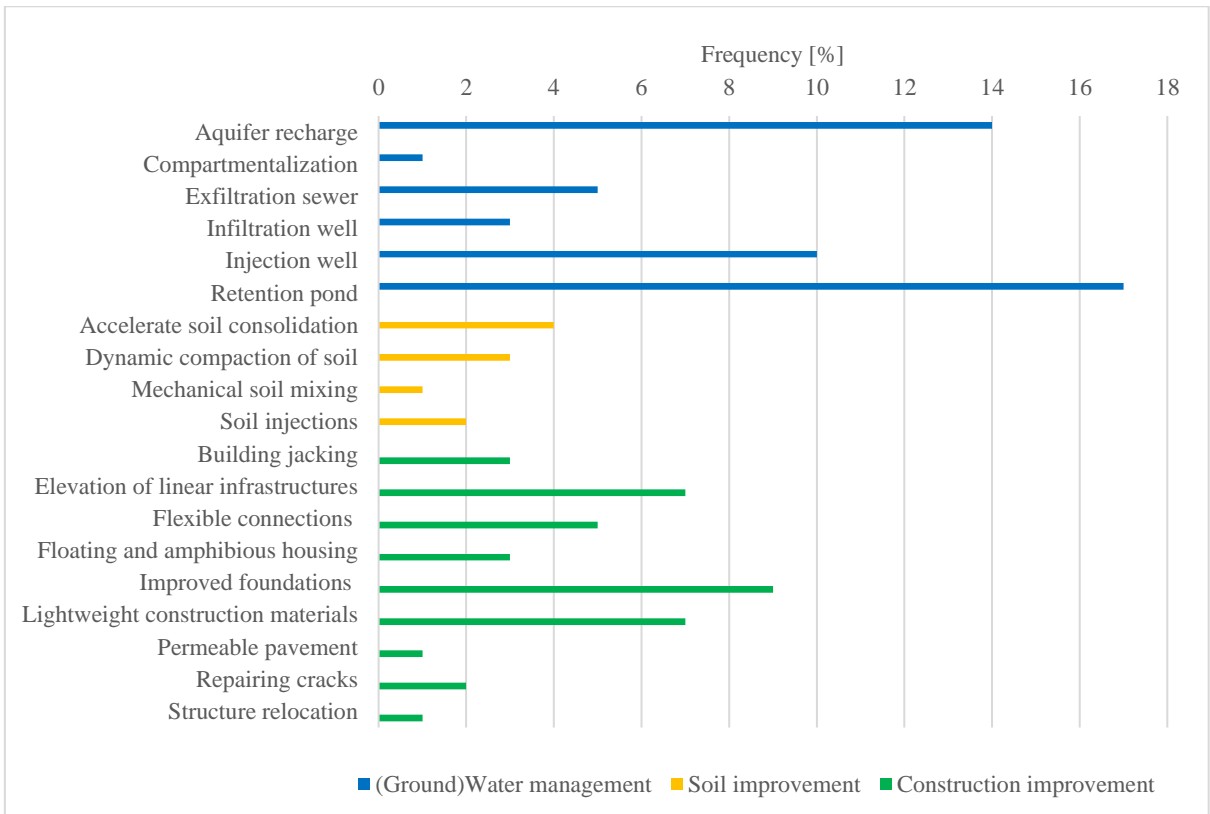

**Figure 4. Frequency distribution of the subsidence reduction measures in the investigated cases.**

Moreover, only 47% of the cases employ a single subsidence countermeasure; instead, 53% use a combination of measures (see also Table 1). Figure 5 shows a network graph where each node represents a subsidence countermeasure, and each link between two nodes indicates at least one case in which the two measures were used together. The subsidence countermeasure with the highest number of connections (11 links) is 'Improved foundations'. Notably, 'Mechanical soil mixing' was used exclusively in combination with 'Accelerate soil consolidation'.





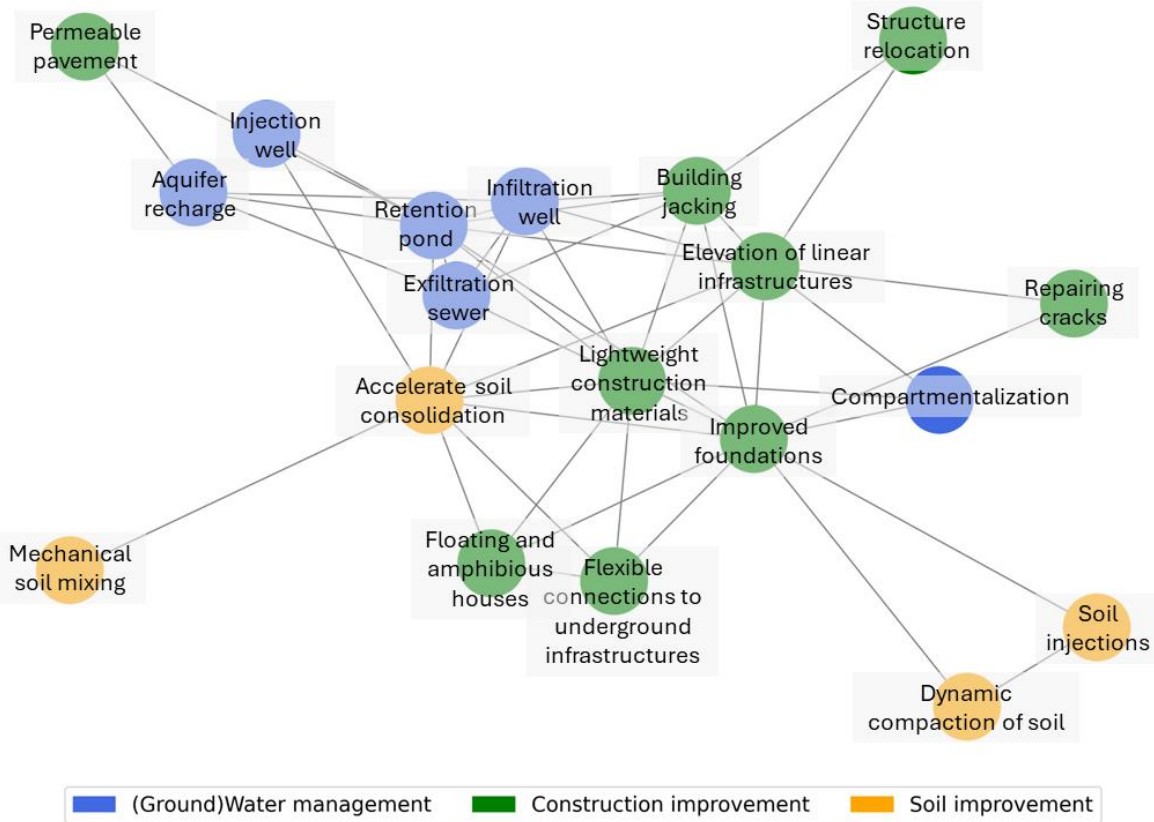

**Figure 5. Network graph illustrating the connections among subsidence reduction measures used in the investigated cases. Each**
**node represents a distinct countermeasure, while the connections between nodes indicate that two corresponding measures were**
**implemented together in at least one of the investigated cases.**

## 5.1 Applicability of subsidence reduction measures

The Question and Response (Q&R) system introduced in Section 4.1 was applied to evaluate the applicability of the subsidence

reduction measures employed in the 49 investigated case studies (see Table B1 in Appendix B). Figure 6 illustrates the average

results per subsidence countermeasure derived from the literature review, expert sessions and surveys. This figure can be used

to identify suitable subsidence reduction measures for a specific case by disregarding those that do not meet the requirements,

which can be done by checking the categories in the columns. Alternatively, the graph can be used to evaluate the applicability

of existing subsidence countermeasures by reading it horizontally along the rows. A square marker indicates that a subsidence

reduction measure belongs to a specific category or that a category includes a particular measure. When a subsidence reduction

measure does not belong to a category, no markers are shown.



**Figure 6. Subsidence reduction measures categorised according to the Question-and-Response (Q&R) system. The squares indicate**
**the association between a measure and a category. The vertical grey shades highlight different groups of categories. Refer to Table**
**B1 in Appendix B for a detailed version.**

**5.2 Effectiveness of subsidence reduction measures**

The four indicators presented in Section 4.2 – reduction potential (RP), operational reliability (OR), negative impact (NI) and
service life (SL) – were applied to evaluate the effectiveness of the subsidence reduction measures adopted in the 49
investigated case studies (see Table B1 in Appendix B). Table 4 summarizes the average results per subsidence countermeasure
based on the outcomes of the literature review, expert sessions and surveys. For some subsidence reduction measures, some



indicators are missing due to insufficient information in the consulted sources. This limitation should be taken into account when using Table 4.


**Table 4. Performance of subsidence reduction measures assessed using four indicators of effectiveness: reduction potential, operational reliability, negative impact and service life. This is a concise version of Table B1 in Appendix B.**

| Subsidence reduction measure | Indicator of effectiveness | | | |
|---|---|---|---|---|
| | Reduction potential | Operational Reliability | Negative impact | Service life |
| *(Ground)Water management* | | | | |
| **Aquifer recharge** | High | Fair | Significant | Long |
| **Compartmentalization** | - | Good | Minimal | Long |
| **Exfiltration sewer** | Medium | Good | Significant | Long |
| **Infiltration well** | High | Good | Significant | Long |
| **Injection well** | Medium | Fair | Significant | Long |
| **Retention pond** | Medium | Good | Significant | Long |
| *Soil improvement* | | | | |
| **Accelerate soil consolidation** | High | Good | Significant | Long |
| **Dynamic compaction of soil** | High | - | Significant | Long |
| **Mechanical soil mixing** | - | Good | Minimal | Long |
| **Soil injections** | Medium | Fair | - | Short |
| *Construction improvement* | | | | |
| **Building jacking** | High | Fair | Significant | Long |
| **Elevation of linear infrastructures** | High | Good | Minimal | Long |
| **Flexible connections** | - | Fair | Minimal | Long |
| **Floating and amphibious houses** | High | Fair | Minimal | Long |
| **Improved foundations** | High | Good | Minimal | Long |
| **Lightweight construction materials** | High | Fair | Minimal | Long |
| **Permeable pavement** | - | Fair | Minimal | Long |
| **Repairing cracks** | High | Good | Minimal | Short |
| **Structure relocation** | High | Good | Significant | Long |



**5.3 Selection of subsidence countermeasures based on applicability and effectiveness**

This section demonstrates the application of the proposed procedure to three well-documented case studies to simulate its use in real-life scenarios.

- *Shanghai (China)*

First reports of subsidence in Shanghai (China) due to groundwater extraction date back to 1921, with an average rate of 2.6
cm/year (Erkens and Stouthamer, 2020; Yang et al., 2020; Ye et al., 2016). The extraction of groundwater for both domestic and industrial use peaked in the 1950s, accelerating subsidence up to 17 cm/year (Gambolati and Teatini, 2021). To contrast the spread of subsidence, restrictions on groundwater extraction were established in the 1960s (Han, 2003; Huang et al., 2015; Shi et al., 2016; Wu et al., 2020). During the same period, a network of extensometers, benchmarks and groundwater observation wells was installed to monitor subsidence (Erkens and Stouthamer, 2020; Ye et al., 2016).

In this context, subsidence countermeasures are necessary to mitigate subsidence in the predominantly soil-based areas of Shanghai at large/regional scale. Based on their applicability (see Section 5.1), four options are suitable: aquifer recharge (surface and trenches), compartmentalization, injection well, and retention pond. Considering their effectiveness (see Section 5.2), this selection can be narrowed down to three options: aquifer recharge (surface and trenches), injection well, and retention pond.

The literature indicates that, due to the topography and land use of the city, injection wells were employed to recharge deep aquifers (Han, 2003; Huang et al., 2015; Shi et al., 2016; Wu et al., 2020). Other measures, such as aquifer recharge from surface, were considered unfeasible due to higher costs (Shi et al., 2016). Nowadays, the monitoring network closely controls the rate of subsidence in Shanghai, maintaining it below 0.6 cm/year (Yang et al., 2020). If subsidence exceeds this threshold, the amount of injected water is adjusted, and additional countermeasures are implemented as necessary (Erkens and
Stouthamer, 2020). The quality of the injected water is also closely monitored to minimize pollution and prevent clogging of pores (Shi et al., 2016).

It can be concluded that the subsidence countermeasure employed in Shanghai in real life aligns with the results of the proposed approach. The final selection among equally viable measures primarily depends on cost considerations and more detailed evaluations that are not part of the current approach.


- *Jakarta (Indonesia)*

Subsidence was first observed in Jakarta (Indonesia) during Dutch colonization in 1925-1926, although little is known about the sinking rates measured at the time (Abidin et al., 2005). In Jakarta, subsidence was slow to be acknowledged as a potential disaster. Investigations were discontinued until 1978, when the impacts of subsidence became evident as cracking of
(infra)structures, lowering of the groundwater level, increased sea water intrusion and expansion of the flood-prone area (Abidin et al., 2011; Andreas et al., 2018; Erkens and Stouthamer, 2020). The first levelling measurements indicated an average



subsidence rate of 6 cm/year, with peaks of 160 cm between 1991 and 1997 (Abidin et al., 2005, 2011). Continuous groundwater extraction, extensive urbanization and the presence of relatively young alluvial soils have since increased subsidence rates, with current velocity of 11-12 cm/year in the most affected areas of Jakarta (Abidin et al., 2015). Only after

a severe flood in 2007 that submerged 40% of the city, local authorities and governments recognized the severity of the problem and began seeking solutions to mitigate and prevent subsidence and damage to structures (Bucx et al., 2015; Erkens et al., 2015).

In this context, a wider range of subsidence countermeasures is applicable to reduce i) subsidence at large/regional scale and ii) damage to structures at small/medium scale (see Section 5.1). Measures to reduce subsidence at large/regional scale include

aquifer recharge (surface and trenches), compartmentalization, injection well, and retention pond. Measures to reduce damage to structures at small/medium scale include building jacking, elevation of linear infrastructures, lightweight construction materials, and structure relocation. Based on their effectiveness (see Section 5.2), compartmentalization and building jacking should be discarded.

According to the literature, to contrast subsidence and damage to structures, regulations on groundwater extraction, building

jacking and elevation of linear infrastructures with sand fill have been extensively adopted in Jakarta (Akbar et al., 2019; Andreas et al., 2018; Saputra et al., 2017, 2019). Additional countermeasures, such as retention ponds, aquifer recharge and exfiltration sewers have been proposed in recent years (Abidin et al., 2015; Akbar et al., 2019; Pramono, 2021). However, the issue in Jakarta is so severe that local governments decided to relocate a consistent portion of the city (Herrera-García et al., 2021).

Similarly to Shanghai, the subsidence countermeasures employed in Jakarta align with the results of the proposed approach. It is interesting to notice how building jacking, which was implemented in real life in Jakarta but discarded by the proposed procedure, proved ineffective in contrasting damage to structures.

- *San Joaquin Valley (USA, California)*

Subsidence in San Joaquin Valley (USA, California) due to groundwater extraction for agriculture was observed since the 1920s (Galloway and Riley, 1999). Continuous exploitation of deep confined aquifer and the consequent compaction of soil led to subsidence of up to 8.53 m. In the 1960s, an extensive monitoring network composed of 31 extensometers was implemented to measure soil compaction rates and determine the extent of subsidence (USGS, 2024). Since the 1970s, alternative surface water, such as the California Aqueduct and other canals, have been supplied allowing a gradual reduction

of groundwater extraction. However, droughts in 1976-77, 1986-92, 2007-09, and 2012-2015 drastically reduced surface water availability, causing a renewed increase in groundwater extraction and aquifer compaction (Galloway and Riley, 1999, USGS, 2024).

Based on their applicability (see Section 5.1), suitable measures to mitigate subsidence in this soil-dominant area on a large/regional scale include aquifer recharge (surface and trenches),  compartmentalization, injection well, and retention pond.



310 Considering their effectiveness (see Section 5.2), this selection can be narrowed down to aquifer recharge (surface and trenches), injection well, and retention pond.

The Sustainable Groundwater Management Act (SGMA) - a legislation passed in 2014 – represents a significant step towards sustainable water management effort in San Joaquin Valley to contrast groundwater depletion, aquifer compaction and droughts (Lees et al., 2021). Nowadays, initiatives have been launched to replenish groundwater by recharging shallow aquifers

315 through surface water percolation, thus helping to balance extraction with natural recharge rates (Lees et al., 2021). Additionally, promoting sustainable water use practices in agriculture and urban areas has become a priority to minimize wastage (USGS, 2024).

Similarly to the previous cases, the subsidence countermeasure employed in San Joaquin Valley aligns with the results of this study. This case further underlines the importance of detailed assessments of the suitability of subsidence reduction measures

320 to also address changing climates and promote sustainable solutions.

## 6 Discussion

In the previous section, a review of 49 cases distributed in 18 countries gathered from scientific papers, technical articles, expert sessions and surveys was conducted to formulate a twofold strategy to select subsidence reduction measures in urban areas based on their applicability and performance. The proposed method consists of two steps: the Question-and-Response

325 (Q&R) system for identifying measures tailored to the specific requirements of each case, and the indicators of effectiveness for evaluating the performance of subsidence countermeasures.

The Q&R system proved useful for an initial screening of subsidence reduction measures. Seven questions were determined to categorize the subsidence countermeasures based on the area's geology, cause of subsidence, scale of application, objective of the intervention and type of urban area. With this system, stakeholders and decision makers can determine the applicability

330 of measures to specific cases and focus on a more limited number of choices. Each subsidence reduction measure can satisfy the requirements of multiple categories, which can be combined to create tailored decision trees. The proposed Q&R system could be further refined by adding sub-categories accounting for location, construction and maintenance costs, hydro-geological, geotechnical and structural engineering settings. Also, the current Q&R system disregards the indirect effects of subsidence (e.g., the increased risk of flooding or seawater intrusion). In a more comprehensive risk management framework,

335 where subsidence is not the only treat, the Q&R system should be improved to account for multiple hazards and effects.

The indicators of effectiveness proposed in this paper (i.e., reduction potential, operational reliability, negative impact and service life) allowed an initial assessment of the performance of subsidence reduction measures. Using these indicators, stakeholders and decision makers can rapidly assess the effectiveness of suitable subsidence reduction measures selected via the Q&R system. Further improvements of the proposed method may involve novel indicators, such as inclusiveness (what

340 societal groups are targeted) and responsibility (allocation of risks in public-private partnerships). At this stage, the proposed procedure allows a qualitative assessment of effectiveness based on the joint evaluation of each indicator's performance. The



evaluation of performance in Table 4 needs further refinement by considering a broader and well-documented range of cases. Currently, the information available to structure the scoring is limited, as demonstrated in Table B1 in Appendix B, and this limitation should be taken into account when applying the indicators from Table 4. This further underlines the need to collect

and share experiences in evaluating the performance of subsidence reduction measures to create a more systematic framework. Once a sufficient number of applications is available for each subsidence reduction measure, quantitative estimations and ranking will also be possible. Additionally, more research is needed to determine the acceptable or unacceptable thresholds for the indicators of effectiveness, also considering the positive or negative interaction of subsidence countermeasures with adjacent assets.

The cases of Shanghai (China), Jakarta (Indonesia) and San Joaquin Valley (USA, California) demonstrate that the proposed two-step procedure to select subsidence countermeasures based on their applicability and effectiveness is promising. In both Shanghai and San Joaquin Valley, where the problem was well-formulated and the key requirements were specific, the Q&R system identified four suitable options, then narrowed down to three by the indicators of effectiveness. In both cases, all the subsidence countermeasures employed in reality were among the proposed options resulting from this procedure. In Jakarta

(Indonesia), eight different options were identified by the system based on their applicability, then narrowed down to six based on their effectiveness. In this case, five subsidence countermeasures employed or proposed in reality were correctly identified by the system, whereas one was discarded.

This demonstrates that, besides the necessary refinements to enhance the accuracy of the proposed method in selecting subsidence reduction measures, careful interpretation of the results is essential. This involves considering the wide variety of

subsidence reduction measures, the causes of subsidence, the site-specific settings and any potential negative or secondary effects. For a thorough validation of the proposed method, a detailed evaluation of effectiveness via measurable parameters – such as water table levels, water infiltration rates, volume of extracted or recharged water, soil compaction, surface rebound, settlement rates, crack widths – is crucial. It is rather surprising how few cases are reported in literature, and even fewer with sufficient evaluation of effectiveness. The consistent use of the four indicators of effectiveness specifically derived for

evaluating the subsidence countermeasures presented in this paper can serve as the basis and catalyst for this.

**7 Conclusions**

Subsidence is a relatively slow process with moderate intensity that is rarely perceived as an imminent disaster. However, its physical, socioeconomic and environmental impacts in urban areas require tailored reduction policies encompassing both mitigation and prevention strategies.

After defining key terminology (i.e., reduction, mitigation, prevention, adaptation, structural and non-structural measures), this paper proposed a twofold strategy to select structural (i.e., technical) measures to contrast subsidence and its physical consequences in urban areas based on their applicability and effectiveness. The objective is to assist stakeholders and decision




makers in managing subsidence (risk) in urban areas, with particular attention to the planning and implementation phases of the subsidence risk frameworks.

Despite the preliminary nature of this work, the proposed methods for selecting subsidence reduction measures and evaluating their effectiveness constitute a novelty in the scientific literature on subsidence studies and mitigation/prevention strategies as no framework currently exists to assess applicable and effective measures. Refinements and further validations are needed to integrate the procedure into current subsidence management practices in urban areas, with specific attention to the local hydrogeological, geotechnical, structural, environmental and social settings where countermeasures are needed. Therefore, at

its current stage, the methodology proposed in this paper should be considered as a preliminary tool for stakeholders and decision makers to identify a set of suitable solutions, which should be further discussed with local experts. Moreover, with appropriate adjustments, the presented methodologies could be applied also for selecting and evaluating the performance of non-structural (i.e., non-technical) measures, subsidence reduction measures in rural areas and secondary subsidence effects.

**Appendix A: Description of subsidence reduction measures**

Table A1 provides a brief description and alternative names of structural (i.e., technical) measures considered in this paper to prevent and mitigate (i.e., reduce) subsidence and its physical consequences in urban areas. The countermeasures in Table A1 are organized in (Ground)Water management, Soil interventions and Construction interventions.

**Table A1. Structural (i.e., technical) measures to reduce subsidence and its physical consequences in urban areas.**

| Subsidence reduction measure | Alternative names | Description |
|---|---|---|
| *(Ground)Water management* | | |
| Aquifer recharge (surface spreading and trenches) | Planned recharge Induced recharge Artificial recharge | Water is spread or impounded on the ground surface, so that it infiltrates through permeable soils (sand or gravel) into an unconfined aquifer. Trenches can also be used to collect runoff water and infiltrate it into the soil. |
| Compartmentalization | | Large polder areas are divided into smaller portions by vertical waterproof barriers, typically made of retaining walls or clay walls. This creates a hydraulic barrier in the subsurface between compartments to maintain a stable groundwater level in each compartment. |
| Exfiltration sewer | Exfiltration trench Perforated pipe Clean water collector Exfiltration pipe | Perforated pipes (usually in PVC or vinyl) redistribute excessive surface or runoff water into the soil while being conveyed. If the groundwater level around the perforated pipe is higher than the water table inside the pipe, then the water conveys as in a conventional sewer. Downpipes from rooftops can be directed the exfiltration sewer instead of wastewater sewers. The |



| | | |
|---|---|---|
| | | exfiltration sewers can be connected to retention ponds and infiltration wells and, if the water needs to be moved from lower to higher altitude, a mechanical water pump can facilitate the circulation of water. |
| Injection well | Recharge well Artificial fluid injection Deep wells | Deep confined aquifers are repressurized by injecting fluids through wells into porous geologic formations (sand, gravel or clay). The injection pipe is usually placed in a fiberglass-reinforced plastic casing. The well is finished with cement grouting, sand, well screen and gravel pack. |
| Infiltration well | Biopore hole | Excessive surface water is collected into a perforated plastic pipe of typically 10 cm in diameter during rainfall events and it is redistributed into compacted soils with poor infiltration rate. The infiltration wells can be also connected to sewer exfiltration systems, and they can be filled with organic waste to improve soil fertilization. |
| Retention pond | Retention basin Catchment area/basin Wet/Storm pond Rainwater harvesting Water banking | This is a permanent catchment area suitable for urban areas to provide additional water storage capacity and attenuate surface runoff during rainfall events. By placing coarse draining material at the bottom (bed) of the pond, water can filtrate in the surrounding soil keeping the desired groundwater level. |
| | | *Soil improvement* |
| Accelerate soil consolidation | | Vertical drains, sand pipes and trenches are placed up to a depth of 35 m to quickly dissipate excessive pore water from soft or organic soils, thus accelerating their consolidation. Additional loads can be applied to the soil by lowering the atmospheric pressure inside the drains, and therefore apply vacuum pressure. This method is usually used to prepare the soil before the construction of (infra)structures. |
| Dynamic compaction of soil | | A heavy steel weight is repeatedly dropped on the ground surface to generate vibrations that, once transmitted to the subsurface, improve and densify soils and filling materials. It is mainly used to treat soils beneath foundations before the construction of (infra)structures. Therefore, the steel weight is dropped in selected locations forming a regular grid pattern. |
| Mechanical soil mixing | Deep soil mixing | Natural soil is mixed with cement or compound binders to improve its mechanical and physical properties. The mechanical binders can be operated in either wet or dry conditions, depending on the typology of soil and the improved characteristics to be achieved. |



| | | |
|---|---|---|
| Soil injections | Void filling<br>Subgrade stabilization | Additives are injected into the subsurface through one or more pipes installed vertically into the ground, thus improving the strength, load-bearing capacity and stability of soft soils. Natural materials as sand, fly ashes or rock powder are mostly used for soft soils. Crushed waste concrete, tire crumb rubber, hydrated lime, resins and polymers have been tested successfully in clay soils. Jet grouting of Portland cement or chemical grouts and foams are mostly used when cavities form into the ground. |
| *Construction improvement* | | |
| Building jacking | Construction lift<br>House raising or lifting | A construction is lifted above its existing foundation to (re-)build a new one at a higher or similar level. |
| Elevation of linear infrastructures | Sand fill | The surface area of infrastructures as roads and railways are lifted by placing an additional layer of material (typically sand and/or road material) on top of existing subsiding layers. In case of bridges, also new (deep) foundations are usually built to elevate the bridge shoulders. |
| Flexible connections to underground infrastructures | Flexible joints | Thermoplastic composite materials or flexible connections are used to join two components of (underground) infrastructures, as pipelines, thus permitting relative movements and providing them with major flexibility. |
| Floating and amphibious houses | | Houses can be built on a water body and be designed with a floating system at their base to allow them floating on water. |
| Improved foundations | Foundation strengthening, replacement, repair, restoration or improvement of foundations | Several methods allows to repair, restore, improve or replace (building) foundations to re-establish their structural capacity:<br><br>• Slab jacking, also called concrete lifting, slab levelling or mud jacking. It is a reparation method used to relevel uneven or sinking concrete slabs. Small holes are drilled into the concrete slab, and strong cementing mixture is injected under the slab to align it back to its original position. Cement mixture, polymer resin, sand, gravel, ash and polyurethane foam can be used as base material.<br>• Underpinning, also called piering. A system of vertical anchors is installed below an existing foundation to reach deeper soil layers with better geo-mechanical properties. This method can be used either to strengthen an existing foundation or to improve the soil before placing a new foundation system. Different techniques can be adopted to achieve this:<br>a. Mass concrete underpinning. The soil around an existing foundation is excavated through controlled stages (or pins) and, when a new suitable foundation soil layer is reached, the excavation is filled with concrete. |



| | | |
|---|---|---|
| | | b. Cantilever needle beam underpinning. The area surrounding the foundation is excavated and a cantilever needle beam is placed through a hole cut in the existing foundation wall. The beam is supported by micropiles, which are placed before excavation. |
| | | c. Pier and beam underpinning. Helical or push piers made of galvanized or epoxy-coated steel are drilled below the foundation till reaching a suitable depth where concrete bases are placed. |
| | | d. Micropiling underpinning. Micropiles are driven below the existing foundation with a certain inclination. Earth is excavated till the top of the pile to be able to replace the earth between the foundation and the pile with concrete. |
| | | e. Pile underpinning. Piles are driven in the proximity of a foundation wall. Then, a needle beam is placed through the foundation wall and connected to the adjacent piles. <ul><li>Installation of (additional) piles. It consists in placing (additional) piles or micro-piles below an existing (shallow) foundation to redistribute the loading.</li><li>Reduction of bacterial decay in wooden piles. Wooden piles area treated with special coatings to preserve them from unforeseen anaerobic conditions and degradation.</li><li>Reduction of negative adhesion/friction around piles. When piles pass through cohesive soils, they can experience negative adhesion due to downwards shear drag movements. This can be reduced by using anti-friction coatings around the piles, by improving the soil characteristics with injections, or by using slender pile sections (e.g., H-pile or precast pile) with smaller pile area.</li><li>Reinforced geotextiles. Geotextiles can be places on top of a system of piles to improve their bearing capacity. This technique is used often to reinforce the foundations of roads and railways.</li></ul> |
| Lightweight construction materials | | Lightweight aggregates can be added to the cement to reduce the construction load. Pumice, scoria, volcanic cinders, tuff, diatomite, heating clay, shale, slate, diatomaceous shale, perlite, obsidian and vermiculite can be used as lightweight aggregates. For road construction, cellular geosynthetics (geofoams and geocombs), the block-moulded expanded polystyrene (EPS) and recycled plastic can be used. |
| Permeable pavement | Permeable paving or porous asphalt | A porous paving surface is made of permeable pavers (in concrete or polymer), concrete or asphalt that allow surface or rainwater to pass through or around them and be slowly infiltrated into the soil. This pavement allows reducing the runoff volume and peak rates of water discharge, and it is mostly used for parking lots, sidewalks or low-traffic roads. |





| Repairing cracks | Different foam- and resin-based materials are used to repair cracks that appear on building facades or road pavements. Additional filling materials are fibre cement, epoxy resin, non-shrink grouts, hot rubber and polymer asphalt. |
|---|---|
| Structure relocation | Buildings are physically moved from their original location to another. This can be done by disassembling and reassembling the construction, or by transporting it whole to the new location. This method is used especially for monumental buildings. |


## Appendix B: Applicability and effectiveness

Table B.1 reports the assessment of applicability (see Section 4.1) and effectiveness (see Section 4.2) of the subsidence reduction measures adopted in the 49 cases investigated in this paper.

**Table B.1. Assessment of applicability and effectiveness of subsidence reduction measures employed in the 49 investigated cases derived from literature review, expert sessions and surveys. The applicability results from the Question-and-Response (Q&R) decision tree system. Effectiveness is evaluated using the indicators of reduction potential (RP), operational reliability (OR), negative impact (NI) and service life (SL). NA denotes 'Not Available' information.**

| Reference | Applicability | | | | | Indicator of effectiveness | | | |
|---|---|---|---|---|---|---|---|---|---|
| | Scale | Objective | Target | Urban area | Space | RP | OR | NI | SL |
| *Aquifer recharge (surface spreading and trenches)* | | | | | | | | | |
| Abidin et al., 2015 | Regional | Mitigation | Hazard, vulnerability & exposure | Rehabilitation | Public, private | NA | NA | NA | NA |
| Bell et al., 2002 | Large | Mitigation | Hazard | Rehabilitation | NA | High | Fair | Significant | Long |
| Han, 2003 | Large, regional | Mitigation | Hazard | Rehabilitation | NA | NA | NA | NA | Long |
| Jha et al., 2009 | Large | Mitigation | Hazard | Rehabilitation | Public | Medium | NA | Significant | Long |
| Nutalaya et al., 1996 | Large | Mitigation | Hazard | Rehabilitation | Public | High | NA | Significant | Long |
| Pacheco-Martínez et al., 2013 | Regional | Mitigation | Hazard, vulnerability & exposure | Rehabilitation | Public, private | NA | Bad | Significant | Long |
| Poland, 1984 | Regional | Mitigation | Hazard | Rehabilitation | NA | NA | NA | NA | NA |
| Sneed and Brandt, 2020 | NA | Mitigation | Hazard | Rehabilitation | Public | NA | NA | NA | NA |





| Szucs et al., 2009 | NA | Mitigation | Hazard | Rehabilitation | Public | NA | NA | Significant | Long |
|---|---|---|---|---|---|---|---|---|---|
| Ting et al., 2020 | Regional | Mitigation | Hazard | Rehabilitation | Public | NA | Good | Minimal | Long |
| Expert sessions and survey | Regional | Mitigation | Hazard | Rehabilitation | Public | NA | NA | NA | NA |
| *Compartmentalization* | | | | | | | | | |
| Kok and Hommes-Slag, 2020 | Large | Prevention, mitigation | Hazard | Rehabilitation | Public, private | NA | Good | Minimal | Long |
| *Exfiltration sewer* | | | | | | | | | |
| Jha et al., 2009 | Medium | Mitigation | Hazard | Rehabilitation | Public | Medium | NA | Minimal | Long |
| McBean et al., 2019 | Medium | Mitigation | Hazard | Rehabilitation | NA | NA | NA | NA | Long |
| Pramono, 2021 | Small | Mitigation | Hazard | Rehabilitation | Private | NA | NA | NA | NA |
| Expert sessions and survey | Medium | Prevention, mitigation | Hazard, vulnerability & exposure | Rehabilitation | Public | High | Good | Significant | Long |
| *Infiltration well* | | | | | | | | | |
| Andriani et al., 2021 | Medium | Prevention, mitigation | Hazard | Rehabilitation | Public | NA | NA | NA | Long |
| Saputra et al., 2017 | NA | Mitigation | Hazard | Rehabilitation | Public | NA | NA | NA | NA |
| Szucs et al., 2009 | NA | Mitigation | Hazard | Rehabilitation | Public | NA | NA | Significant | Long |
| Expert sessions and survey | Small | Prevention, mitigation | Hazard, vulnerability & exposure | Rehabilitation | Public | High | Good | Significant | Long |
| *Injection well* | | | | | | | | | |
| Brighenti, 1991 | NA | Mitigation | Hazard | NA | NA | NA | NA | Significant | Short |
| Galloway and Riley, 1999 | Regional | Mitigation | Hazard | Rehabilitation | Public | NA | NA | Significant | Long |
| Gambolati et al., 2005 | Regional | Mitigation | Hazard | Rehabilitation | Public | Medium | NA | Minimal | Short |
| Han, 2003 | Regional | Mitigation | Hazard | Rehabilitation | NA | NA | NA | NA | Long |
| Huang et al., 2015 | Medium | Mitigation | Hazard | Rehabilitation | Public | NA | NA | Minimal | Long |
| Li et al., 2021 | Regional | Mitigation | Hazard | Rehabilitation | NA | NA | NA | Significant | Long |
| Phien-Wej et al., 1998 | Medium, large | Mitigation | Hazard | Rehabilitation | Public | NA | NA | Significant | Short |



| Poland, 1984 | Regional | Mitigation | Hazard | Rehabilitation | NA | NA | NA | Significant | Long |
|---|---|---|---|---|---|---|---|---|---|
| Shi et al., 2016 | Regional | Mitigation | Hazard | Rehabilitation | Public | NA | Fair | Significant | Long |
| Tang et al., 2022 | Regional | Mitigation | Hazard | Rehabilitation | Public | NA | NA | Minimal | Long |
| Testa, 1991 | NA | Mitigation | Hazard | NA | NA | Low | NA | NA | NA |
| Wu et al., 2020 | Regional | Mitigation | Hazard | Rehabilitation | Public | NA | NA | Minimal | Short |
| Yang et al., 2020 | NA | Mitigation | Hazard | Rehabilitation | Public | NA | Good | Minimal | Long |
| Ye et al., 2016 | Regional | Mitigation | Hazard | Rehabilitation | NA | NA | NA | Minimal | Long |
| Expert sessions and survey | Medium | Prevention, mitigation | Hazard | Rehabilitation | Public | Medium | Fair | Significant | Short |
| *Retention pond* | | | | | | | | | |
| Akbar et al., 2019 | Large | Mitigation | Hazard | Rehabilitation | Public | NA | NA | NA | NA |
| Andriani et al., 2021 | Large | Mitigation | Hazard | Rehabilitation | Public | NA | NA | NA | Long |
| Bell et al., 2002 | Large | Mitigation | Hazard | Rehabilitation | NA | High | Fair | Significant | Long |
| Galloway and Riley, 1999 | Large, regional | Mitigation | Hazard | Rehabilitation | Public | NA | NA | Significant | Long |
| Han, 2003 | Regional | Mitigation | Hazard | Rehabilitation | NA | NA | NA | NA | Long |
| Jha et al., 2009 | Large | Mitigation | Hazard | Rehabilitation | Public | Medium | - | Significant | Long |
| Lixin et al., 2022 | Regional | Prevention, mitigation | Hazard | Rehabilitation | Public | NA | NA | NA | NA |
| Poland, 1984 | Regional | Mitigation | Hazard | Rehabilitation | NA | NA | NA | NA | Long |
| Pramono, 2021 | Regional | Mitigation | Hazard | Rehabilitation | Public | NA | NA | NA | NA |
| Sneed and Brandt, 2020 | NA | Mitigation | Hazard | Rehabilitation | Public | NA | NA | NA | NA |
| Szucs et al., 2009 | NA | Mitigation | Hazard | Rehabilitation | Public | NA | NA | Significant | Long |
| Ting et al., 2020 | Regional | Mitigation | Hazard | Rehabilitation | Public | NA | Good | Minimal | Long |
| Zektser et al., 2005 | NA | Mitigation | Hazard | Rehabilitation | NA | NA | NA | NA | Long |
| Expert sessions and survey | Large | Mitigation | Hazard | Rehabilitation, new development | Public | Medium | Good | Significant | Long |
| *Accelerate soil consolidation* | | | | | | | | | |
| Andriani et al., 2021 | Regional | Prevention | Hazard | Rehabilitation, new development | Public, private | NA | NA | NA | Long |
| Bergado et al., 1993 | Medium | NA | Hazard | NA | NA | NA | NA | NA | Long |





| Poland, 1984 | NA | Mitigation | Hazard | Rehabilitation | NA | NA | Good | NA | Long |
|---|---|---|---|---|---|---|---|---|---|
| Ritzema, 2015 | Medium | Prevention, mitigation | Hazard | New development | Public, private | NA | NA | NA | NA |
| Expert sessions and survey | Medium | Prevention | Hazard | New development | Public, private | High | Good | Significant | Long |
| *Dynamic compaction of soil* | | | | | | | | | |
| Al-Zabedy and Al-Kifae, 2020 | Large | Prevention | Hazard | New development | NA | High | NA | NA | Long |
| Hamidi et al., 2011 | Medium, large | Prevention | Hazard | New development | Public, private | NA | NA | Significant | Long |
| Liang et al., 2015 | Medium | Mitigation | Hazard | Rehabilitation | Public, private | NA | NA | Significant | Long |
| Shen et al., 2019 | Medium | Mitigation | Hazard | Rehabilitation | NA | NA | NA | NA | Long |
| *Mechanical soil mixing* | | | | | | | | | |
| Bergado et al., 1993 | Medium | Mitigation | Hazard | Rehabilitation | Public | NA | Good | Minimal | Long |
| *Soil injections* | | | | | | | | | |
| Al-Zabedy and Al-Kifae, 2020 | Large | Prevention | Hazard | New development | NA | Medium | NA | NA | Long |
| Xuan et al., 2015 | Medium | Prevention | Hazard | New development | NA | High | Fair | NA | Short |
| *Building jacking* | | | | | | | | | |
| Andreas et al., 2018 | Small, Medium | Prevention | Vulnerability & exposure | Rehabilitation | Public, private | NA | NA | NA | NA |
| Saputra et al., 2017 | NA | Mitigation | Vulnerability & exposure | Rehabilitation | Private | NA | NA | NA | NA |
| Expert sessions and survey | Micro, small | Prevention | Vulnerability & exposure | Rehabilitation | Public, private | High | Fair | Significant | Short |
| *Elevation of linear infrastructures* | | | | | | | | | |
| Akbar et al., 2019 | Medium | Mitigation | Vulnerability & exposure | Rehabilitation | Public, private | NA | NA | NA | NA |
| Andreas et al., 2018 | Micro, small, Medium | Prevention, mitigation | Vulnerability & exposure | Rehabilitation | Public | NA | NA | NA | NA |
| Andriani et al., 2021 | Medium | Mitigation | Hazard | Rehabilitation | Public | NA | NA | NA | Long |



| | | | | | | | | | |
|---|---|---|---|---|---|---|---|---|---|
| Carreón-Freyre et al., 2010 | Small, medium | Prevention, mitigation | Vulnerability & exposure | Rehabilitation | Public | High | Good | Minimal | Long |
| Kok and Hommes-Slag, 2020 | Large | Prevention, mitigation | Vulnerability & exposure | Rehabilitation | Public, private | NA | NA | NA | NA |
| Poland, 1984 | Medium | Prevention, mitigation | Vulnerability & exposure | Rehabilitation | NA | NA | NA | NA | Long |
| *Flexible connections to underground infrastructures* | | | | | | | | | |
| Alferink and Cordóva, 2017 | Micro, small | Prevention | Vulnerability & exposure | Rehabilitation, new development | NA | NA | Fair | Minimal | Long |
| Gutiérrez and Cooper, 2002 | Micro | Prevention, mitigation | Vulnerability & exposure | Rehabilitation, new development | NA | NA | NA | NA | NA |
| Paukstys et al., 1999 | Small | Prevention | Vulnerability & exposure | New development | NA | NA | NA | NA | NA |
| Ritzema, 2015 | Small | Prevention, mitigation | Vulnerability & exposure | Rehabilitation, new development | Public, private | NA | NA | NA | NA |
| *Floating and amphibious housing* | | | | | | | | | |
| Basak and Chowdhury, 2021 | Medium | Prevention | Vulnerability & exposure | New development | Private | NA | NA | Minimal | Long |
| English et al., 2016 | Medium | Prevention | Vulnerability & exposure | New development | Private | NA | NA | Minimal | Long |
| Pötz and Bleuzé, 2009 | Medium | NA | NA | NA | NA | NA | NA | NA | Long |
| Ritzema, 2015 | Medium | Prevention | Vulnerability & exposure | New development | Private | NA | NA | NA | NA |
| Expert sessions and survey | Medium | Prevention | Vulnerability & exposure | New development | Private | High | Fair | Minimal | Long |
| *Improved foundations* | | | | | | | | | |
| Al-Zabedy and Al-Kifae, 2020 | Regional | Prevention | Vulnerability & exposure | New development | NA | Medium | NA | NA | Long |



| Deakin, 2005 | Micro, small | Mitigation | Vulnerability & exposure | Rehabilitation | Private | NA | NA | NA | Short |
|---|---|---|---|---|---|---|---|---|---|
| Gutiérrez and Cooper, 2002 | Micro, small | Prevention, mitigation | Vulnerability & exposure | Rehabilitation, new development | NA | NA | NA | NA | NA |
| Kok and Hommes-Slag, 2020 | Large | Prevention, mitigation | Vulnerability & exposure | Rehabilitation | Private | NA | NA | Minimal | Long |
| Ovando-Shelley et al., 2013 | NA | Prevention, mitigation | Vulnerability & exposure | Rehabilitation | Public, private | NA | Good | Minimal | Long |
| Poland, 1984 | Large | Mitigation | Vulnerability & exposure | Rehabilitation | NA | NA | NA | NA | Long |
| Ritzema, 2015 | Medium | Prevention, mitigation | Vulnerability & exposure | Rehabilitation, new development | Public, private | NA | NA | NA | NA |
| Expert sessions and survey | Small, medium | Prevention, mitigation | Vulnerability & exposure | Rehabilitation | Private | High | Good | Minimal | Long |
| *Lightweight construction materials* | | | | | | | | | |
| Andriani et al., 2021 | Medium | Mitigation | Hazard | Rehabilitation, new development | Public, private | NA | NA | NA | Long |
| Kohlnhofer, 1992 | NA | Prevention | Vulnerability & exposure | New development | Public | NA | NA | Minimal | Long |
| Kok and Hommes-Slag, 2020 | Large | Mitigation | Vulnerability & exposure | Rehabilitation | Public, private | NA | NA | NA | NA |
| Ritzema, 2015 | Medium | Prevention | Hazard, vulnerability & exposure | New development | Public, private | NA | NA | NA | NA |
| Saputra et al., 2017 | NA | Prevention, mitigation | Vulnerability & exposure | Rehabilitation | NA | NA | NA | NA | NA |





| Expert sessions and survey | Small, medium | Prevention, mitigation | Vulnerability & exposure | Rehabilitation, new development | Public, private | High | Fair | Minimal | Long |
|---|---|---|---|---|---|---|---|---|---|
| *Permeable pavement* | | | | | | | | | |
| Poland, 1984 | Medium | Mitigation | Hazard | Rehabilitation | NA | NA | Fair | Minimal | Long |
| *Repairing cracks* | | | | | | | | | |
| Carreón-Freyre et al., 2010 | Micro | Mitigation | Vulnerability & exposure | Rehabilitation | Public | High | Good | Minimal | Long |
| Deakin, 2005 | Micro, small | Mitigation | Vulnerability & exposure | Rehabilitation | Private | NA | NA | NA | Short |
| Luo et al., 2019 | Micro | Mitigation | Vulnerability & exposure | Rehabilitation | Public | NA | NA | NA | Short |
| *Structure relocation* | | | | | | | | | |
| Andreas et al., 2018 | Micro, small, large | Mitigation | Vulnerability & exposure | New development | Public, private | High | Good | Significant | Long |

**Author contribution**

NN and MK conceptualized the research. NN collected and analysed the data, designed the methodology and prepared the manuscript – draft and edited version. MK reviewed the manuscript and supervised the activities involved in the research.

**Competing interests**

The authors declare that they have no conflict of interest.

**Acknowledgments**

The research presented in this article is part of the project Living on soft soils: subsidence and society (grantnr.: NWA.1160.18.259), which is funded by the Dutch Research Council (NWO-NWA-ORC), Utrecht University, Wageningen University, Delft University of Technology, Ministry of Infrastructure & Water Management, Ministry of the Interior & Kingdom Relations, Deltares, Wageningen Environmental Research, TNO-Geological Survey of The Netherlands, STOWA, Water Authority: Hoogheemraadschap de Stichtse Rijnlanden, Water Authority: Drents Overijsselse Delta, Province of Utrecht, Province of Zuid-Holland, Municipality of Gouda, Platform Soft Soil, Sweco, Tauw BV, NAM.



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
