# Peer review of "Applicability and effectiveness of structural measures for subsidence (risk) reduction in urban areas"

_EGUsphere, 2024_

## Author Response (AR1)

**Response to Reviewers' Comments**

**We sincerely thank the reviewers for their insightful comments and suggestions, which have significantly improved the quality of our manuscript. Below, we provide a point-by-point response to each comment, detailing the changes made to the manuscript. Our responses are highlighted in bold red.**

**RC1: 'Comment on egusphere-2024-2537', Samar Momin, 18 Nov 2024**

**General Comments:**

The paper presents a novel and systematic approach for addressing subsidence risks in urban areas, combining the **Question-and-Response (Q&R)** system with performance indicators to identify and evaluate structural mitigation measures. The methodology's application to global case studies, including Shanghai, Jakarta, and San Joaquin Valley, demonstrates its practical utility. The integration of expert surveys, literature reviews, and real-world data enriches the analysis.

The manuscript is well-organized, with a logical flow that facilitates understanding. Key concepts are defined early, and the proposed methodology is clearly explained. The difference in the use of the terms Structural and Non-structural from those in Structural Engineering applications provides clarity. The findings provide valuable insights for decision-makers, engineers, and urban planners addressing subsidence in diverse geotechnical and socio-economic contexts.

**Strengths:**

1. **Comprehensive Framework**:

   o The twofold strategy, combining the Q&R system with effectiveness indicators, fills a critical gap in subsidence risk management literature.

2. **Broad Application**:

   o The analysis spans 49 cases across 18 countries, providing a robust validation of the methodology.

3. **Clarity in Presentation**:

   o Figures, tables, and case study descriptions are detailed and informative, supporting the narrative effectively.

**Response to Reviewer RC1:**

**We sincerely thank the Reviewer for their evaluation and valuable comments on our manuscript. We improved our manuscript incorporating your suggestions, fixed any inconsistency and double checked the citation formatting.**

**Specific Comments:**

1. **Definitions of Key Terms**:

   o The definitions of "mitigation," "prevention," and "adaptation" are clear. However, these could be linked to more concrete examples from the 49 cases to further ground theoretical explanations.

*Response to Reviewer RC1: The definitions of terms in Section 2 of the manuscript include few examples of mitigation, prevention and adaptation measures. In our view, extending the list of examples at this stage would detract from the clarity of the definitions. Instead, the link between mitigation/prevention definitions and countermeasures adopted in the investigated 49 cases is provided in Section 5.1 "Applicability of subsidence reduction measures", Figure 6. Furthermore, Table B1 in Appendix B offers more specific links between the measures cited in each reviewed paper/case and the objective of the corresponding intervention.*

2. **Limitations of Current Data**:

   o The paper acknowledges that some indicators lack sufficient data. It would benefit from emphasizing the potential bias this introduces and suggesting how future studies could address these gaps.

*Response to Reviewer RC1: In the revised manuscript, we emphasized that "Currently, the information available to structure the scoring is limited, as demonstrated in Table B1 in Appendix B. This affects the criteria used to assign scores, possibly leading to over- or underestimations of the effectiveness of certain subsidence countermeasures. This limitation should be taken into account when applying the indicators from Table 4, as it may influence the selection of subsidence reduction measures. The lack of comprehensive and consistent data further underlines the need to collect and share experiences in evaluating the performance of subsidence reduction measures to create a more systematic framework." To fill this gap, future works should report more systematically the performance of subsidence countermeasures using the indicators of effectiveness presented in this paper, which "can serve as the basis and catalyst for this improvement".*

3. **Comparative Effectiveness**:

   o While the Q&R system identifies suitable measures, the discussion could delve deeper into why certain measures perform better in specific contexts (e.g., why aquifer recharge was prioritized over injection wells in certain cases).

*Response to Reviewer RC1: In the revised manuscript we clarified the methodology used to evaluate the effectiveness of subsidence countermeasures. Specifically, "the effectiveness of subsidence countermeasures is evaluated by assigning equal weight to all indicators of effectiveness, treating them as equally important. The qualitative values of each indicator are scored on a scale from 1 to 3 [...]. The overall effectiveness of subsidence reduction measures is determined by averaging the scores across all indicators." Additionally, when two subsidence countermeasures have equal effectiveness, other factors (such as costs and long-term sustainability) can influence the choice of the*

preferred measure. For example, in Shanghai, injection wells were preferred over aquifer recharge due to their lower costs; while in San Joaquin Valley, aquifer recharge was favored because it allows greater volume of water to be replenished compared to injection wells. These example underline the importance of "*multi-criteria assessments that weigh effectiveness alongside costs and sustainability*", as underlined in the revised Discussion.

4. **Broader Impacts**:

   o The paper lightly touches on the indirect consequences of subsidence measures, such as their environmental and social impacts. Expanding this discussion would strengthen the study's utility for policymakers.

Response to Reviewer RC1: Thank you for the valuable comment. In the revised manuscript, we expanded the discussion on the indirect consequences of subsidence reduction measures, focusing on their environmental, social and economic impacts, which we believe are critical for policymakers. Particularly, we added that "*Environmental considerations encompass potential alterations of local ecosystems, changes to water quality, depletion of natural resources, and increased energy consumption with associated carbon emissions. For instance, creating artificial retention ponds may disrupt natural habitats, while recharging aquifers – whether through surface or deep injections – may affect water quality. Similarly, infrastructure-heavy solutions, such as injection wells, can contribute to greenhouse gas emissions, especially if they rely on non-renewable energy sources. Social impacts involve the displacement of communities due to relocation, leading to social stress and loss of community identity. Equity issues can also arise, as the increased costs or reduced availability of potable water disproportionately affect low-income populations. Other economic implications include rising property values, which can lead to gentrification and displacement of lower-income residents. Given these challenges, stakeholder and decision makers should adopt a multidimensional approach to subsidence (risk) management that integrates technical considerations with environmental stewardship, social equity, and economic feasibility. Proactively addressing the indirect consequences of subsidence reduction measures can contribute to sustainable and resilient urban development.*"

**Technical Comments:**

1. **Grammar and Style**:

   o Minor grammatical inconsistencies were noted, such as:

      ▪ Line 263: "If subsidence exceeds this threshold" could clarify what "this threshold" specifically refers to.

Response to Reviewer RC1: Thank you for the suggestion. In the revised version of the manuscript we change "*If subsidence exceeds this threshold*" to "*If subsidence exceeds 0.6 cm/year*".

2. **Citations and Formatting**:

   o Some citations lack consistent formatting. For instance, references to multiple authors (e.g., Akbar et al., 2019; Herrera-García et al., 2021) are not uniform across the text.

**Response to Reviewer RC1: Thank you for pointing this out. We checked the citation format across the test and fixed any inconsistency.**

**RC2**: 'Comment on egusphere-2024-2537', Anonymous Referee #2, 19 Nov 2024

Nappo and Korff have presented a novel framework for evaluating the applicability and effectiveness of different subsidence countermeasures derived from existing literature. Except the exclusion of cost in their framework, I believe that it fills a major gap in the ability to undertake a structured approach to mitigating subsidence impacts.

A major concern for me is the authors' approach to concluding the validity of their methodology based on 3 example studies. I do not have the same confidence as them in their conclusions due to data leakage and over-reliance on one category - size of the impacted region. As a suggestion for improving this part of the manuscript, instead of using the example studies as evidence for their framework's validity, they could be used as motivating and detailed examples for developing the framework.

**Response to Reviewer RC2:**

**We sincerely thank the Reviewer for their evaluation and valuable comments on our manuscript. We carefully revised our manuscript addressing your suggestions and resolving any inconsistency. And we believe that our revised manuscript now aligns with your expectations. Below we provide a response to your questions and comments.**

1. The authors have not included cost of a countermeasure in their Q&R system. This is a major gap and limitation of their system as cost is often a primary factor in determining the appropriate countermeasure.

**Response to Reviewer RC2: We agree that the cost of subsidence countermeasures is a relevant factor in practice when selecting suitable options. We acknowledge that costs should be considered as sub-category in the Q&R system. However, to facilitate a comparison among worldwide case studies, "this paper does not include any sub-category" such as costs, as it "would hinder the comparability of diverse applications", as explained in Section 4.1. In the revised manuscript, we also expanded the Discussion to highlight the importance of economic considerations in decision-making. We also recognize that the integration of cost considerations into the framework is an important step for future refinements, and we propose exploring this in future work. This would allow for a more comprehensive decision-making process that accounts for both technical effectiveness and financial feasibility.**

2. The Q&R system has been presented as a decision tree approach, however, decision trees were not employed in the 3 example study regions in the manuscript.

**Response to Reviewer RC2: We appreciate you comment and appreciate the opportunity to clarify the application of Q&R system to the three examples. The Q&R system is indeed designed as a decision tree, though it was not formally illustrated as such in the manuscript. By progressively answering the questions outlined in Section 4, a user can identify a set of applicable subsidence countermeasures for a given case study, taking into account the geology, primary cause of subsidence, desired intervention, targeted action, scale of application, and type of urban setting. This same process was applied to the three case studies (Shanghai, Jakarta, and San Joaquin Valley), which demonstrate how the Q&R**

**system can be used in practice. The selection of subsidence countermeasures in each case derives from the comprehensive responses to all the questions in the system. We hope this explanation clarifies the intended use of the Q&R system as a decision tree approach in the case studies.**

3. While the example regions used by the authors are geographically diverse, their subsidence category of large/regional scale is the same. In order to draw conclusions about the effectiveness of authors' methodology, a more diverse set of example cases should have been reviewed.

**Response to Reviewer RC2: We acknowledge the Reviewer's concern regarding the scale of application in the selected case studies. To address this, in the revised manuscript we reclassified the scale of application of subsidence countermeasures using a logarithmic scale. This allows a better diversification of scales.**

4. The authors have used existing literature to populate the results of their Q&R system in Figure 6. Then the system has been applied to 3 of the regions from the same literature and presented as evidence of the validity of the authors' methodology. As a result of this data leakage, I don't believe that the conclusions on the effectiveness of the authors' methodology can be drawn here.

**Response to Reviewer RC2: We understand your concerns regarding the validity of the methodology based on the three examples. As clarified in Section 5.3, the cases of Shanghai, Jakarta and San Joaquin Valley were selected from the total of 49 reviewed case studies because they are the most extensively documented, either through available literature or the insights provided by academic scholars and experts on subsidence mitigation and prevention. While these cases rely on existing literature, the Q&R system's framework is designed to generalize across various scenarios by systematically evaluating factors such as geology, causes of subsidence, and urban settings. Unfortunately, no other cases had such detailed information on the history of subsidence management to be included as additional examples. To further acknowledge this, we remarked in the revised Discussion that additional testing on independent data sets is necessary to fully validate the methodology.**

5. In Section 5, the results are presented as a percentage of the 49 cases. While percentages can be useful in larger datasets, for a small dataset, I would suggest additionally including the counts in the results.

**Response to Reviewer RC2: We changed the text accordingly in the revised manuscript.**

6. Line 32 - The sentence would be clearer by separately classifying groundwater, for example, groundwater extraction...

**Response to Reviewer RC2: We changed the text as suggested in the revised manuscript.**

7. Line 36 - *Unlike*

**Response to Reviewer RC2: We changed the text as suggested in the revised manuscript.**

8. Line 110 - Please cite Table 1 for this sentence.

**Response to Reviewer RC2: We changed the text as suggested in the revised manuscript.**

9. Line 149 - Could the authors clarify what metric is used for measuring reduction - is it subsidence risk that would include the hazard along with exposure and vulnerability?

**Response to Reviewer RC2: We appreciate the opportunity to clarify this point. The reduction potential (RP) indicator focuses specifically on the physical reduction of subsidence (i.e., diminished settlement) and damage (e.g., reduced width or number of cracks). The RP is assessed by comparing the amount of subsidence or damage observed before and after the implementation of a subsidence countermeasure. We clarified this in the revised manuscript to ensure that the definition of RP is clear.**

10. Line 154 - Could the authors clarify if "time" refers to the entire service life of the countermeasure?

**Response to Reviewer RC2: In the revised manuscript, we clarified that operational reliability refers to the functionality of subsidence countermeasures during their service life.**

11. Line 164 - *causes*

**Response to Reviewer RC2: We changed the text as suggested in the revised manuscript.**

12. Line 164 - Could the authors provide examples of detrimental impacts?

**Response to Reviewer RC2: In the revised manuscript we provided examples of detrimental effects "e.g., water pollution, pore clogging, increase of subsidence, (increased) damage to adjacent structures".**

13. Line 171 - Could the authors elaborate on how 10 years was selected as the serviceable life threshold, as it appears quite low for structural countermeasures? Typically most non-critical structures are engineered for 50 years, and critical structures for longer than that.

**Response to Reviewer RC2: Thank you for this valuable suggestion. We carefully revised the manuscript and updated the classification of service life (SL) to better reflect typical service life of engineering structures (50 – 100 years). The revised thresholds are defined as follows: short (SL< 20 years), medium (SF: 20-50 years) and long (SF> 50 years).**

14. Line 184 - 32% of the 49 cases *have* a secondary cause. "Also" is not necessary given the "additionally".

**Response to Reviewer RC2: We changed the manuscript as suggested.**

15. Fig 2 - *Tertiary*

**Response to Reviewer RC2: We changed the text in Figure 2 as suggested.**

16. Fig 2 - Since the authors have mentioned "anthropogenic and natural" in the figure caption, it would be clearer if these were also identified in the figure.

**Response to Reviewer RC2: We incorporated this suggestion in the revised Figure 2.**

17. Fig 3 caption - geological types *of* the investigated

**Response to Reviewer RC2: We changed the caption of Figure 3 as suggested.**

18. Fig 5 - It is an impressive figure that makes it very easy to see which measures are used in conjunction. As a minor suggestion, the figure could be further improved by assigning edge weight based on the number of times that edge is present in the case studies.

**Response to Reviewer RC2: We revised Figure 5 as suggested. It now shows different edges based on the number of times that edge is present in the case studies.**

19. Section 5.2 - The authors have mentioned the number of experts as 13 elsewhere but it will also be helpful to mention it in this section.

**Response to Reviewer RC2: We modified the text as suggested.**

20. Section 5.2 - Did the number of occurrences of a particular countermeasure across the 49 case studies influence the expert responses? How did the authors ensure that frequency of countermeasure did not influence its performance assessment?

**Response to Reviewer RC2: Thank you for raising this important point. The frequency of subsidence countermeasures in the 49 case studies did not influence the expert responses, as the academic scholars and experts were unaware of this information during the expert sessions and surveys. However, it is important to note that the effectiveness of each countermeasure is assessed based on the mode, meaning that the frequency of a countermeasure does influence the performance assessment. We discuss this limitation in the revised Discussion. Specifically, the frequency of countermeasures may lead to over- or underestimations of their effectiveness. "This limitation should be taken into account when applying the indicators from Table 4, as it may influence the selection of subsidence reduction measures."**

21. Line 236 - How is the average computed? Given the qualitative nature of all responses, it would not be ideal if 2 responses of High and Low are averaged as medium. Rather, mode may be a better metric.

**Response to Reviewer RC2: We agree with the Reviewer regarding the use of the mode as a more appropriate metric. Indeed, in the revised manuscript we clarified that "The mode is used as metric to assign a single value to each indicator of effectiveness for the subsidence countermeasures. In cases of equally frequent results, expert judgment is preferred if available; otherwise, the highest value is assigned. It is important to note that for some subsidence reduction measures some indicators are missing due to insufficient information in the consulted sources (see Table B1 in Appendix B). When no data is available, no value is assigned to the corresponding indicator of effectiveness."**

22. Line 238 - What is the definition of "insufficient information"? What is the minimum number of responses required to populate each cell in Table 4?

**Response to Reviewer RC2: As shown in Table B1 in Appendix B, the minimum number of responses required to populate each cell in Table 4 is 1. We believe that a stricter criterion on the response frequency would have limited the number of conclusions that could be drawn. However, we recognize this as a limitation, which is discussed further in the manuscript. Moreover, we advise additional testing on independent data sets and "a detailed evaluation of effectiveness via measurable parameters – such as water table levels, water infiltration rates, volume of extracted or recharged water, soil compaction, surface rebound, settlement rates, crack widths".**

23. Line 255 - It would be helpful to elaborate why mitigation is needed over a "large/regional scale", for example, by providing the area of subsidence region.

**Response to Reviewer RC2: Thank you for your valuable comments. In the revised manuscript we provided additional details on the Shanghai case study by specifying the extent of the area affected by subsidence (i.e., more than 90,000 km$^2$ ), where countermeasures are needed. This further clarify why the "regional scale" is used.**

24. Line 257 - It is not clear which effectiveness parameters led to removal of compartmentalization as a countermeasure.

**Response to Reviewer RC2: We clarified in the revised manuscript that "The effectiveness of subsidence countermeasures is evaluated by assigning equal weight to all indicators of effectiveness, treating them as equally important. The qualitative values of each indicator are scored on a scale from 1 to 3 [...]. The overall effectiveness of subsidence reduction measures is determined by averaging the scores across all indicators. This approach ensures a balanced evaluation of all criteria and facilitates the prioritization of subsidence countermeasures." Typically, the top two subsidence reduction measures are considered the most applicable and effective.**

25. Line 267 - It is not clear how the authors reached their conclusion. They only used one category - regional scale - for their analysis and did not consider any of the other categories. Additionally, as the authors highlight on Line 268 that viability primarily depends on cost considerations, why was cost of a countermeasure excluded from their Q&R system?

**Response to Reviewer RC2: We acknowledge that the size of the impacted region is an important factor, but we would like to emphasize that the choice of subsidence countermeasures is based on the comprehensive responses to all questions in the Q&R system, not just this one category. Additionally, while we acknowledge that the viability of subsidence countermeasures, particularly on a regional scale, often depends on cost considerations, we intentionally excluded cost from the Q&R system to facilitate a broader comparison across different regions. Including cost as a sub-category could have hindered the comparability of diverse applications, as discussed in Section 4.1. However, we highlight the importance of considering cost and other factors like sustainability in the Discussion.**

26. Line 275 - Wouldn't "lowering of the groundwater level" be a cause of subsidence instead of its impact?

**Response to Reviewer RC2: Thank you for pointing this out. In the revised manuscript, we clarified that "*excessive extraction of groundwater caused the water table to drop significantly, limiting the access to clean potable water*".**

27. Line 277 - Does the peak refer to 160 cm/year subsidence or total subsidence of 160 cm over some time period, at a particular site?

**Response to Reviewer RC2: In the revised manuscript we clarified that "between 1991 and 1997, the cumulative subsidence reached up to 160 cm, particularly in the costal areas".**

28. Line 283 - It would be helpful to elaborate why and how the two regional scales are determined, for example, by providing the extent of subsidence.

**Response to Reviewer RC2: Thank you for your valuable comments. In the revised manuscript, we provide additional details on the Jakarta case study, which helped clarifying the scale of application for the subsidence countermeasures. As part of the revisions, we now use a logarithmic scale for categorizing the scale of application, which reclassifies the area of Jakarta – which is 660 km² – as large scale, rather than regional as previously described. This reclassification not only differentiates Jakarta from Shanghai, but also led to a reassessment of the applicability of countermeasures.**

29. Line 287 - It is not clear which effectiveness parameters led to removal of compartmentalization and building jacking as countermeasures.

**Response to Reviewer RC2: In the revised manuscript, the reclassification of the scale of application – which now uses a logarithmic scale – led to a reassessment of the applicability of countermeasures. Therefore, compartmentalization is no longer an option in this case.**

**Some countermeasures, such as building jacking, were not selected in our analysis because they were considered less effective that other options. The qualitative values of each indicator are scored on a scale from 1 to 3 [...]. The overall effectiveness of subsidence reduction measures is determined by averaging the scores across all indicators. This approach ensures a balanced evaluation of all criteria and facilitates the prioritization of subsidence countermeasures." Typically, the top two subsidence reduction measures are considered the most applicable and effective.**

30. Line 295 - Similar to my previous comment for Shanghai, I am not sure if the author's conclusion can be effectively drawn from their limited analysis. Especially as 2 (building jacking, exfiltration sewers) out of 6 implemented or proposed countermeasures are not among the authors' recommendations.

**Response to Reviewer RC2: Thank you for pointing this out. As clarified in the revised manuscript, measures as aquifer recharge, injection wells and exfiltration sewers were implemented in Jakarta largely in response to freshwater scarcity and flood risk, rather than subsidence reduction. Similarly, *"retention ponds have been constructed to manage rainwater runoff and reduce flooding in the Kebon Jeruk sub-district"*. These measures were therefore not considered as options to reduce subsidence in our methodology.**

31. Line 296 - I was unable to find any supporting information in the manuscript for the ineffectiveness of building jacking in Jakarta.

**Response to Reviewer RC2: As clarified in the revised manuscript, building jacking was not selected in our analysis because considered less effective that other options. The qualitative values of each indicator are scored on a scale from 1 to 3 [...]. The overall effectiveness of subsidence reduction measures is determined by averaging the scores across all indicators. This approach ensures a balanced evaluation of all criteria and facilitates the prioritization of subsidence countermeasures." Typically, the top two subsidence reduction measures are considered the most applicable and effective.**

32. Line 303 - What is the time period over which the subsidence of 8.53 m was observed? Additionally, was it observed over a broad area or was this the peak at a single site?

**Response to Reviewer RC2: In the revised manuscript, we clarified that "*Continuous exploitation of deep confined aquifer and the consequent soil compaction caused an area larger than 10.000 km$^2$ to sink by an average of 31 cm between 1925 and 1970 (Galloway and Riley, 1999). In some localised areas, subsidence reached up to 8.53 m during the same period"*.**

33. Line 308 - Similar to my previous comments, please provide supporting evidence for selecting large/regional scale; and the determination of ineffectiveness of compartmentalization.

**Response to Reviewer RC2: Thank you for the valuable comments. In the revised manuscript we provided additional details on the San Joaquin Valley case study, such as the extent of the area affected by subsidence (i.e., 10.000 km² ) and the sinking rate of 31 cm between 1925 and 1970. The new classification of the scale of application - which now uses a logarithmic scale – allowed to assign better the scale of application to each measure. These changes determined that compartmentalization is considered suitable for medium scales, which is not the case of San Joaquin Valley.**

34. Line 318 - Similar to my previous comments, the conclusion drawn by the authors seems quite subjective as 1 (sustainable water use practices) of 2 implemented countermeasures was not part of their suggestions.

**Response to Reviewer RC2: In San Joaquin Valley, the two options resulting from our procedure were either implemented (aquifer recharge) or considered as an alternative (injection wells). The final choice of subsidence countermeasure was influenced by additional considerations, such as the volume of recharged water and the sustainability of the measures. However, these considerations are not part of our methodology in its current form, though they should be considered for future improvements, as highlighted in the Discussion.**

35. Line 350 - It is not clear to me why the authors' methodology could only be evaluated for the 3 example studies, since all the countermeasures presented in the manuscript and their applicability in Figure 6 are derived from existing literature.

**Response to Reviewer RC2: We appreciate the opportunity to clarify this point. While the subsidence countermeasures presented in the manuscript and their applicability are derived from existing literature, the cases of Shanghai, Jakarta and San Joaquin Valley were chosen to demonstrate how the proposed procedure can be used in practice. These examples were selected from the total of 49 reviewed case studies because they are the most extensively documented. We have updated the manuscript to emphasize this perspective, which better reflects the preliminary nature of this work.**